# Breathing coordinates cortico-hippocampal dynamics in mice during offline states

Nikolaos Karalis [1,2✉] & Anton Sirota [1✉]

Network dynamics have been proposed as a mechanistic substrate for the information transfer across cortical and hippocampal circuits. However, little is known about the mechanisms that synchronize and coordinate these processes across widespread brain regions during offline states. Here we address the hypothesis that breathing acts as an oscillatory pacemaker, persistently coupling distributed brain circuit dynamics. Using large-scale recordings from a number of cortical and subcortical brain regions in behaving mice, we uncover the presence of an intracerebral respiratory corollary discharge, that modulates neural activity across these circuits. During offline states, the respiratory modulation underlies the coupling of hippocampal sharp-wave ripples and cortical DOWN/UP state transitions, which mediates systems memory consolidation. These results highlight breathing, a perennial brain rhythm, as an oscillatory scaffold for the functional coordination of the limbic circuit that supports the segregation and integration of information flow across neuronal networks during offline states.

[1] Faculty of Medicine, Ludwig-Maximilian University, Munich 82152 Martinsried, Germany. [2] Friedrich Miescher Institute for Biomedical Research, 4058 Basel, Switzerland. ✉email: nikolaskaralis@gmail.com; sirota@bio.lmu.de

Over the past century, cortical and subcortical structures of the limbic circuit and the medial temporal lobe have been identified as critical elements of the memory circuit, involved in the formation and retrieval of episodic memories during online states[1,2] and their consolidation during sleep[3–5]. Despite a growing understanding of local plasticity rules and neural activity correlates systems-level mechanisms that enable processing and transfer of information across distributed circuits are not well understood.

Global, slow neuronal oscillations have been proposed as a mechanism that enables the coordinated interaction between remote brain regions, serving as a source of synchronization in the local circuits and promoting synaptic plasticity[6,7]. During active behavioral states, sensory-motor integration loops engage widespread circuits and give rise to theta oscillations that entrain local dynamics and provide the substrate for the coordination of the information flow and population coding[5,6,8,9]. In contrast, during slow-wave sleep, the cortex is in a bistable state, characterized by alternations between generalized silent DOWN, and synchronous UP states across thalamocortical circuits[5,7], while the hippocampal circuits transiently synchronize during sharp-wave bursts associated with high-frequency ripple oscillations[10]. These bistable dynamics are pair-wise coordinated[11–15], giving rise to emergent spatiotemporal neural population dynamics that reactivate awake memory patterns[16–20], while their interaction has been postulated to support memory consolidation[11,13,21–23] and the transfer of memories to their permanent cortical storage[24,25].

During offline brain states (such as sleep), the cortex is sensory disconnected from the environment. Systems consolidation across distributed circuits has to rely on the global coupling of internal network dynamics, to enable the coordinated reactivation of previous experiences across remote brain regions. However, the mechanisms that support the coherence of such coordinated dynamics across distributed cortical and subcortical circuits during sleep and quiescence remain unknown. From a theoretical perspective, a global pacemaker has been postulated as an effective solution to the coupling of distinct network dynamics[26–28], but the neural implementation of such a mechanism remains elusive.

Here we address the hypothesis that breathing serves as a pacemaker that couples neuronal dynamics across the limbic system during offline states and supports coordinated information flow across cortico-hippocampal circuits. Using multi-regional recordings of local field potentials (LFP) and large-scale neural population activity we performed an anatomically-resolved, in vivo, functional dissection of the offline state dynamics in the medial prefrontal cortex (mPFC), hippocampus, basolateral amygdala (BLA), nucleus accumbens (NAc), visual cortex, and thalamic nuclei. Using this approach, combined with pharmacological manipulation, we describe the widespread respiratory entrainment of the limbic circuit and we suggest the existence of an intracerebral centrifugal respiratory corollary discharge that mediates the inter-regional synchronization of the limbic circuit.

## Results

### Respiratory entrainment of prefrontal cortex across brain states.
To investigate the role of breathing in organizing neuronal dynamics in the mPFC during offline states, we recorded simultaneously the local electrical activity (electroolfactogram; olfactory EEG)[29] of the olfactory sensory neurons (OSNs), and single-units and LFP in the mPFC in freely-behaving mice during different vigilance states (Fig. 1a and Supplementary Fig. 9a, b). The olfactory EEG reflected the respiratory activity and exhibited a

reliable phase relationship to the respiratory cycle, as established by comparing this signal to the airflow from the nostrils (Supplementary Fig. 1a–d), and was reflected in rhythmic head-motion (Supplementary Fig. 1b). Using the density of head micromotions and muscle twitches, we classified behavioral segments as active awakening, quiescence, or sleep Supplementary Fig. 9i, j) (see Methods). Distinct states are associated with changes in the breathing frequency (Fig. 1c).

Examination of the spectrotemporal characteristics of the respiratory activity and the prefrontal LFP revealed a faithful reflection of the respiratory activity in the prefrontal LFP, characterized by a prominent peak in the 2–6 Hz range (Fig. 1b–e), suggesting a potential relationship between these oscillations. The two oscillations were comodulated across a wide frequency range (Fig. 1f) and this relationship was preserved throughout many active (online) as well as inactive (offline) states in freely-behaving mice (Fig. 1c–e and Supplementary Fig. 1g, h). Coherence and Granger causality analysis of the respiratory and LFP signals suggested that the respiratory oscillation is tightly locked and likely causally involved in the generation of the prefrontal slow-wave LFP oscillation signal during offline states (Fig. 1g–i and Supplementary Fig. 1g, h).

Beyond disengaged offline states, a prominent 4 Hz oscillation is also dominating the mouse mPFC during fear behavior[30]. The similarity in frequency suggests that breathing is the origin of fear-related 4 Hz oscillations. To explicitly test this, we exposed mice to auditory, contextual, and innate fear paradigms (Supplementary Fig. 2a, b). During freezing, the respiratory rhythm changed in a stereotypic manner and matched the rhythmic head-motion and the prefrontal LFP oscillation power, phase, and onset (Supplementary Figs. 2c–h, 8j). Consistent with the results during offline states, the prefrontal LFP oscillation was coherent with and Granger-caused by breathing (Supplementary Fig. 2g–h). Interestingly, the peak breathing frequency was slightly distinct for different types of fear behavior and the prefrontal peak frequency faithfully matched it, in support of the generality of the described phenomenon (Supplementary Fig. 2e).

Having established the generality of the coupling between breathing and LFP oscillation in the mPFC across distinct states, consistent with previous reports[31–33], we focused on the enigmatic and least understood quiescence and slow-wave sleep states. Slow waves in cortical LFP during offline states are linked to the thalamocortical slow oscillation[7,34], therefore the dominance of the respiratory rhythm on the prefrontal LFP warrants careful examination of the role of respiration in offline state dynamics.

To further investigate the extent of respiratory entrainment of prefrontal circuits at the level of neural population activity, we examined the firing of extracellularly recorded single neurons in mPFC in relation to the respiratory phase (Fig. 2a). To achieve that, in addition to the wire-electrode recordings in freely-behaving mice, we performed large-scale silicon probe recordings from head-fixed mice (Supplementary Fig. 9c, d). This configuration enables effective recording from a large neuronal population and across multiple experimental conditions. For the detection of quiescence and sleep periods for head-fixed recordings, we relied on video tracking of the mouse snout and body, from which we derived a micromotion signal, similar to the freely-behaving case. We observed that ~60% of putative prefrontal principal cells (PN) and ~90% of putative inhibitory interneurons (IN), identified based on their extracellular waveform features[35] (Supplementary Fig. 3a, b), were significantly modulated by the phase of the breathing cycle (Fig. 2b, c and Supplementary Fig. 3c) and more strongly modulated by the past phase of the respiratory cycle (Fig. 2i, j). Most modulated cells fired preferentially in the trough/ascending phase of the local

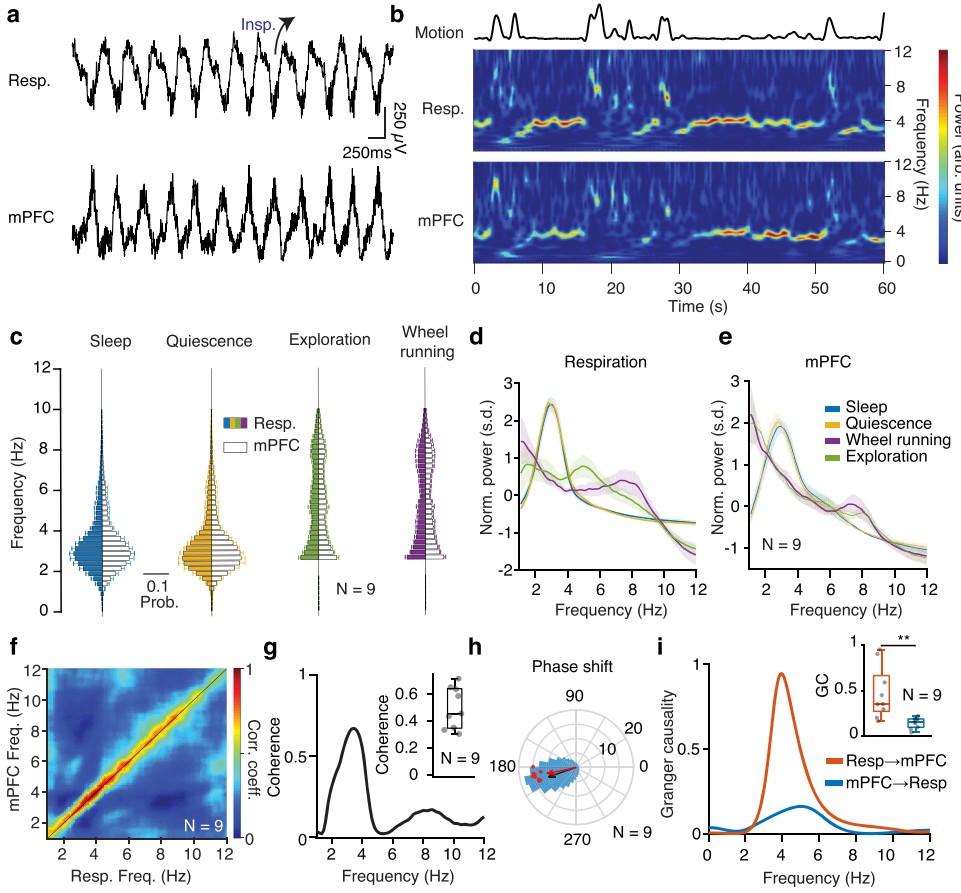

**Fig. 1 Prefrontal oscillations are related to breathing throughout behavioral states. a** Example traces of simultaneously recorded respiratory olfactory EEG and medial prefrontal local field potentials (LFP) in freely-behaving mice. **b** Example time-frequency decomposition of respiratory and mPFC LFP signals, revealing the reliable relationship between the two signals. Top, the motion amplitude of the mouse at each time point. **c** Distribution of peak frequency bins of the spectrally decomposed respiration (left; darker colors) and mPFC LFP (right; lighter colors) during slow-wave sleep, quiescence, exploratory behavior, and self-initiated wheel running (N = 9 freely-behaving mice). **d, e** Averaged normalized power spectral density of respiration (**d**) and mPFC LFP (**e**) across states as in (**c**). **f** Frequency-resolved comodulation of respiration and mPFC LFP oscillation power, across mice and behaviors (N = 9 freely-behaving mice). **g** Example coherence spectrum between respiration and mPFC LFP during offline states. Inset, average coherence value in the 2–5 Hz band (N = 9 freely-behaving mice). **h** Phase shift of 2–5 Hz filtered respiration and mPFC LFP signals during offline states for an example animal (blue histogram) and overlaid magnitude of phase modulation (logZ) and average phase shift for all animals (red dots; N = 9 freely-behaving mice). Black arrow depicts the average phase and logZ of the phase shift for the example and the red arrow for the population. **i** Example spectral Granger causality between respiration and mPFC LFP for both causal directions. Inset, group statistics of the average Granger causality for the 4 Hz band (2–5 Hz) between respiration and mPFC LFP for both causality directions (N = 9 freely-behaving mice, Wilcoxon signed-rank two-sided test, resp → mPFC versus mPFC → resp, P = 0.0039, **P < 0.01). For box plots, the middle, bottom, and top lines correspond to the median, bottom, and top quartile, and whiskers to lower and upper extremes minus bottom quartile and top quartile, respectively. arb. units arbitrary units, s.d. standard deviations. Shaded areas, mean ± SEM. Source data are provided as a Source Data file.

oscillation, corresponding to the inhalation phase (Fig. 2b, c and Supplementary Fig. 3d), largely independently of the ongoing breathing cycle duration (Supplementary Fig. 3g), and are even more strongly modulated by the phase of the local oscillation (Fig. 2c and Supplementary Fig. 3e, f). To ensure that this modulation is not a result of mechanical stimulation of the neurons due to breathing-related brain pulsation in relation to the electrode, we posited that such an artifact would be expected to be most severe for neurons located closer to the electrode, as they would be more likely to be mechanically stimulated. However, we did not find a correlation between modulation strength and distance to the electrode inferred from the amplitude of the action potential or any modulation of the unit amplitude by the respiratory phase (Supplementary Fig. 3i–l).

In order to characterize the effect of this respiratory entrainment on prefrontal population dynamics, we performed dimensionality-reduction (Isomap and PCA) on the correlation

matrix of inspiration-triggered unit activity. Using this approach, we identified that breathing entrains prefrontal population activity in a low-dimensional, periodic orbit, thus exhibiting cyclic attractor characteristics, whereby population dynamics would remain stable across consecutive breathing cycles (Fig. 2d and Supplementary Fig. 4c, d).

**Topography of respiratory entrainment.** The mPFC consists of multiple subregions along the dorsoventral axis, all of which are characterized by differential afferent and efferent connectivity and behavioral correlates[36,37]. To understand the origin and anatomical substrate of the respiratory entrainment of prefrontal circuits, we performed a three-dimensional trans-laminar and trans-regional characterization of the mPFC field potentials using custom-designed high-density silicon probes in head-fixed mice (Fig. 2e–p and Supplementary Fig. 4e). These recordings revealed a consistent increase from dorsal to ventral mPFC in both the

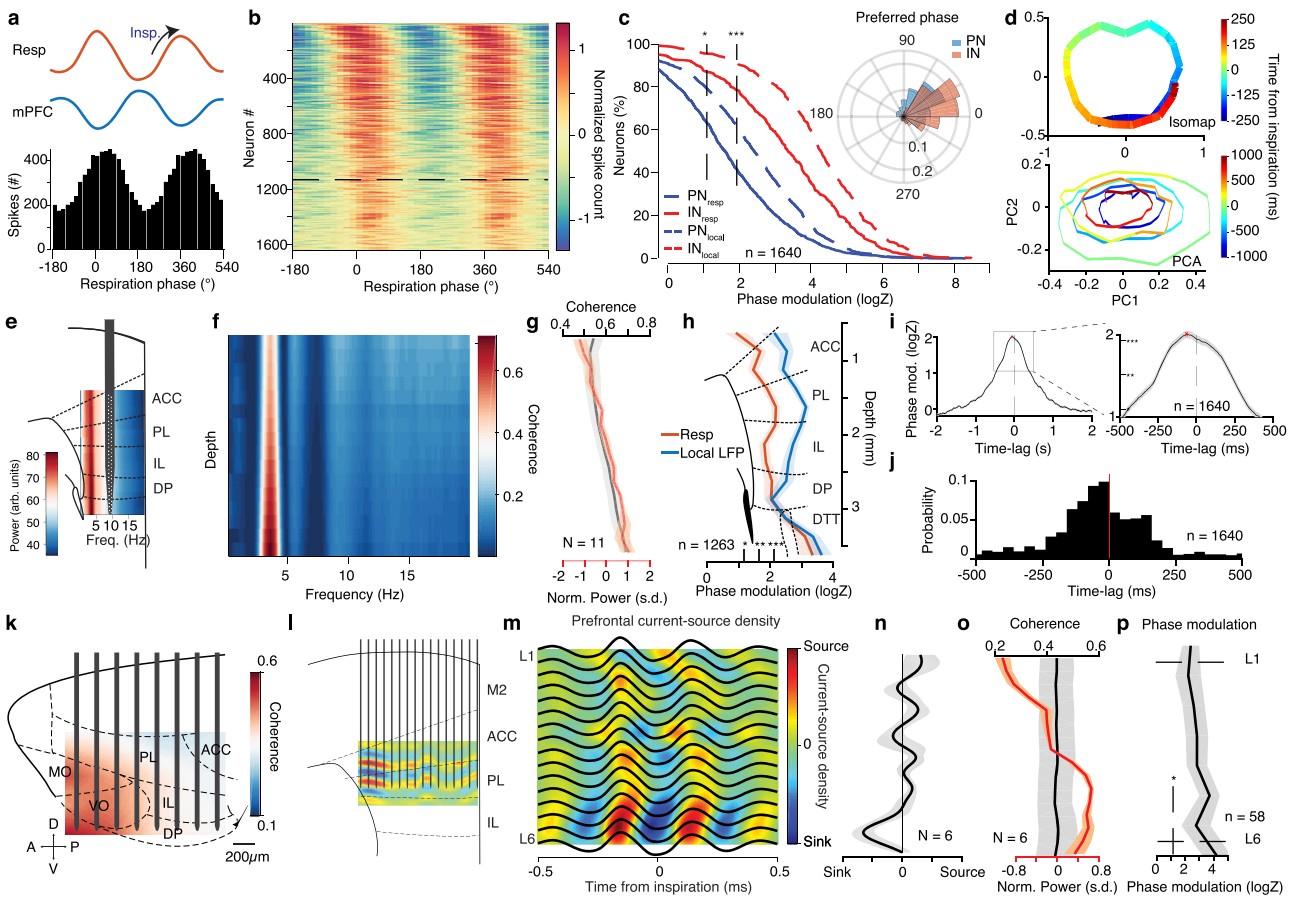

**Fig. 2 Topography of prefrontal circuit entrainment by breathing. a** Example respiration phase histogram. Top, average respiration (red) and mPFC LFP (blue) traces. **b** Color-coded phase histograms of all prefrontal neurons ($n = 1640$ neurons, $N = 13$ mice). Horizontal line: significance threshold. **c** Cumulative distribution of the logZ for all prefrontal PNs (blue, $n = 1250$ neurons) and INs (red, $n = 390$ neurons) in relation to respiration (solid lines) and local LFP (vertical dashed lines). Inset: Distribution of the preferred phase for PNs (blue) and INs (red). **d** Example Isomap (top) and PCA (bottom) trajectories of the mPFC population activity triggered on inspiration. **e** Schematic of a silicon polytrode in the deep layers of the mPFC, overlaid on an example depth- and frequency-resolved power spectrum spanning all mPFC subregions. **f** Example depth- and frequency-resolved coherence between respiration and local prefrontal LFP. **g** Average depth-resolved power (red) and coherence in the 2–5 Hz band (black) ($N = 11$ mice). **h** Depth-resolved spike phase modulation statistics (logZ) (as in (**c**)) ($n = 1263$ cells, $N = 11$ mice). **i** Time-lagged phase modulation of prefrontal population, indicative of directionality from respiration to prefrontal activity. Peak lag is $-65$ms. **j** Time-lag distribution for the prefrontal population ($n = 1640$ neurons, $N = 13$ mice). **k** Example 2D coherence throughout the frontal subregions. **l** Schematic of a translaminar recording from the mPFC using a 16-shank probe (50 μm spacing) and example inspiration-triggered CSD profile. **m** Example average inspiration-triggered LFP traces and overlaid corresponding translaminar CSD from the mPFC. **n** Average inspiration-triggered translaminar normalized CSD from the mPFC ($N = 6$). **o** Average laminar profile of the 2–5 Hz band LFP power (red) and coherence with respiration (coherence) ($N = 6$). **p** Laminar unit phase modulation statistics (logZ) ($n = 58$ cells, $N = 6$). Shaded areas, mean ± SEM, arb. units arbitrary units, s.d. standard deviations, Insp inspiration, ACC anterior cingulate, PL prelimbic, IL infralimbic, DP dorsal peduncular, MO medial orbital, VO ventral orbital, L1 layer 1, L6 layer 6. Stars indicate significance levels (*$P < 0.05$; **$P < 0.01$; ***$P < 0.001$).

power of the breathing-related oscillation and its coherence with breathing (Fig. 2e–g). Similarly, the coherence was stronger in the anterior prefrontal regions (Fig. 2k).

The presence of an oscillation in the mPFC with this particular profile could also be consistent with a volume-conducted signal generated by bulbar dipoles, since olfactory bulb (OB) LFP is dominated by breathing-related oscillations[38–42]. Further, the multitude of reafferent pathways to the prefrontal cortex originating in the OB and their complex 3D axonal termination patterns, combined with the spatiotemporal superposition of multiple laminae-specific dipoles and the volume conduction from distant sites make the exact interpretation of LFP patterns alone challenging. To distinguish these potential sources of origin and identify local current dipoles that might underlie observed prefrontal LFP patterns, we recorded LFP activity across the prefrontal cortical layers and calculated the current-source density (CSD) (Fig. 2l–n and Supplementary Fig. 4b). This

analysis revealed a prominent pattern of sinks in the deep cortical layers at the inspiration phase, giving rise to an increased LFP power and unit-LFP coupling in the deep layers (Fig. 2o, p), weighing against the hypothesis of volume conduction and suggestive of a synaptic origin of the prefrontal LFP oscillation. To further control for the possibility of strong volume-conducted signals affecting the recorded and calculated signals, we constructed a simple model of ventral or lateral oscillatory sources that volume conduct to the recorded region. In both these cases, the model failed to account for the experimentally observed LFP and CSD patterns (Supplementary Fig. 4a).

Harnessing the advantages of spatial information from the silicon probe recordings in head-fixed mice, we characterized the entrainment of single-units across prefrontal subregions and cortical layers during quiescence. Interestingly, cells were phase-modulated throughout the prefrontal subregions (Fig. 2h and Supplementary Fig. 3e, f). These results, given the increased

density of polysynaptic projections from the OB to ventral mPFC subregions[31], suggest that the bulbar reafferent input to the mPFC is giving rise to the observed LFP signals (Supplementary Fig. 3m). Rhythmic airflow could entrain the olfactory sensory neurons (OSNs), that are known to respond both to odors and mechanical stimuli[43], and propagate through the olfactory bulb and the olfactory system to the prefrontal region. To understand the potential role of breathing in orchestrating the hippocampo–cortical dialog, we undertook a detailed investigation of the mechanism and tested for a possible modulatory role of the breathing rhythm on brain circuits.

**Widespread respiratory modulation of limbic circuits during offline states**. Given the prominent modulation of mPFC by breathing, we hypothesized that concurrent entrainment of other regions could be underlying their generalized long-range interaction, as has been suggested before for theta oscillations during active states[9,44]. For this, we turned our attention to other brain regions, reciprocally connected to the mPFC, that are known to interact with prefrontal networks in different behaviors and be involved in memory consolidation[14,45,46].

Using large-scale single-unit and laminar LFP recordings from the dorsal hippocampus in head-fixed mice, we identified that in both dorsal CA1 and dentate gyrus (DG), ~60% of PNs and 80% of CA1 INs were modulated by the phase of breathing during quiescence, firing preferentially after the inspiration (Fig. 3a–c), in line with previous reports of respiratory entrainment of hippocampal activity during immobility and anesthesia[32,47–49]. A separation of CA1 PNs based on their relative position within the pyramidal layer into populations with known distinct connectivity patterns[50,51] did not reveal particular differences in their modulation by breathing, suggesting a generality of this entrainment throughout the CA1 sub-populations (Supplementary Fig. 5d).

To understand what afferent pathways are responsible for breathing-related synaptic currents that underlie the modulation of spiking activity, we calculated the finely-resolved (23 μm resolution) laminar profile of inspiration-triggered dorsal hippocampal CSD, enabling the identification of synaptic inputs into dendritic sub-compartments. Although the LFP profile only highlights the prominence of the respiratory band in the DG hilus region (Supplementary Fig. 5a), depth-resolved coherence analysis identifies coherence with breathing in the dendritic CA1 and DG layers (Supplementary Fig. 5b, c), while high-resolution CSD analysis revealed more accurately the presence of two distinct and time-shifted respiratory-related inputs to DG dendritic sub-compartments (Fig. 3d, e and Supplementary Fig. 5e). Inspiration was associated with an early sink in the outer molecular layer of DG, indicative of input from the layer II (LII) of the lateral entorhinal cortex (LEC), followed by a sink in the middle molecular layer of DG, indicative of input from the layer II of the medial entorhinal cortex (MEC)[52,53] (Fig. 3d, e and Supplementary Fig. 5e). The inputs from MEC and LEC are the two primary cortical inputs to the hippocampus, providing primarily spatial and sensory information respectively[54–56]. The modulation of these inputs by breathing reinforces the notion that the hippocampal respiratory modulation is mediated by entorhinal inputs, while the distinct timing of these two inputs suggests that the respiratory phase might serve as a reference for the temporal organization of the incoming information.

To explore the extent of cortical and subcortical entrainment by breathing, we further recorded LFP and single-unit activity in the BLA, NAc, V1, as well as somatic and midline thalamus in head-fixed mice (Fig. 3f–s and Supplementary Figs. 4e, 9l). Similar to mPFC, during quiescence LFP in both BLA and NAc

was comodulated with breathing across a range of frequencies, with most prominent modulation at ~4 Hz, the main mode of breathing frequency during quiescence (Fig. 3f–i), and exhibited reliable cycle-to-cycle phase relationship with the respiratory oscillation (Supplementary Fig. 6a, b). Given the nuclear nature and lack of lamination of these structures, which obfuscates the interpretation of slow LFP oscillations, we examined the modulation of single-unit activity by the phase of breathing, revealing that a large proportion of BLA, NAc, and thalamic neurons were significantly modulated by respiration, notably firing in distinct phases of the breathing cycle (Fig. 3j–n). In the visual cortex, both LFPs were coherent to (Fig. 3p) and a large fraction of neurons (Fig. 3q) were phase-modulated by breathing. Interestingly, the magnitude of phase modulation was maximal in layers 4–6, which is consistent with both the phase modulation of sensory thalamic neurons (Fig. 3r), the presence of strong coherence of the LFP with respiration in the thalamus (Supplementary Fig. 5b) and respiration-related CSD sinks in the L4/5 where thalamic inputs arrive (Fig. 3s).

**Reafferent origin of limbic gamma oscillations**. A prominent feature of prefrontal LFP is the presence of fast gamma oscillations (~80 Hz), believed to emerge due to local synchronization[57,58] (Fig. 4a). To investigate the relationship of prefrontal gamma oscillations to the breathing rhythm and well-known OB gamma oscillations[40,59–61], we recorded simultaneously from the two structures in head-fixed mice and calculated the phase-amplitude coupling between breathing and gamma oscillations during quiescence (Fig. 4a, b and Supplementary Fig. 6c, d). Both OB and mPFC fast gamma oscillations were modulated by the phase of breathing, such that gamma bursts occurred predominantly simultaneously and in the descending phase of the local LFP (Fig. 4c). The simultaneously occurring OB and mPFC gamma oscillations matched in frequency and exhibited a reliable phase relationship with a phase lag that suggests directionality from the OB to the mPFC (Fig. 4d and Supplementary Fig. 6d). To examine the underlying synaptic inputs mediating the occurrence of these oscillations in the mPFC, we calculated CSD across mPFC layers, triggered on the phase of the OB gamma bursts. This analysis revealed a discrete set of sinks across prefrontal layers associated with OB bursts (Fig. 4i, j). Similar to the slow timescale LFP signals, these results suggest that fast gamma oscillatory signals in mPFC are generated by OB-gamma rhythmic polysynaptic inputs to mPFC and are not exclusively attributable to a locally generated rhythm.

Examination of the OB ~80 Hz gamma-triggered dorsal hippocampal CSD revealed a DG outer molecular layer sink (Fig. 4k, l), a profile distinct from the similar frequency CA1$_{lm}$ gamma (Supplementary Fig. 6e). This would be consistent with LEC LII inputs, a region known to exhibit olfactory-related activity[62]. In parallel, slow BLA gamma (~40 Hz) and fast NAc gamma (~80 Hz) oscillations were modulated by the phase of breathing, occurring predominantly in the trough and ascending phase of breathing respectively (Fig. 4m–o).

To examine whether these breathing-modulated OB-mediated gamma oscillations have a functional role in driving local neuronal activity, we quantified the coupling of local single-units to mPFC gamma signals, revealing that ~40% of principal cells and ~55% of interneurons increased their firing rate in response to local gamma oscillations (Fig. 4e, f). Interestingly, ~10% of PN and ~30% of IN were significantly phase modulated by gamma oscillations, firing preferentially in the trough of the local oscillation (Fig. 4g, h). Similarly, ~40% of BLA and ~25% of NAc cells fired preferentially in the trough of the local gamma oscillations (Fig. 4p and Supplementary Fig. 6f, g), suggesting the

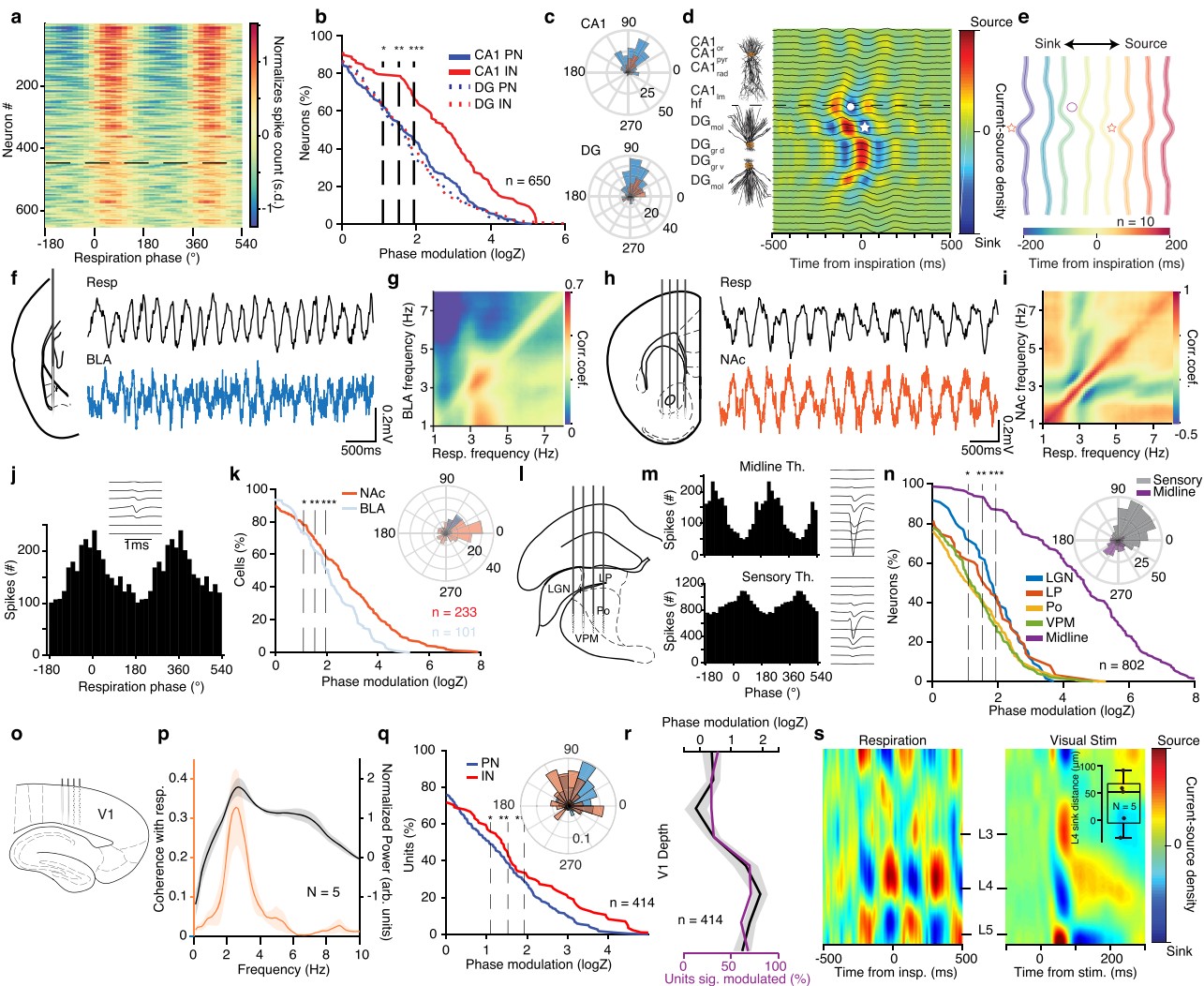

**Fig. 3 Breathing modulates hippocampal, thalamic, BLA, NAc, and V1 neuronal activity. a** Color-coded respiration phase histograms of all hippocampal cells ($n = 650$ cells). Horizontal line: significance threshold. **b** Cumulative distribution of the modulation strength for all CA1 and DG PNs (CA1, $n = 226$ cells; DG, $n = 206$ cells; $N = 22$ mice) and INs (CA1, $n = 98$ cells; DG, $n = 120$ cells). Vertical lines: significance thresholds. **c** Preferred phase distribution for all significantly phase-modulated CA1 and DG cells. **d** CA1 pyramidal and DG granular cell somatodendritic domains aligned to an example inspiration-triggered high-density hippocampal CSD. Line: fissure. **e** Inspiration-triggered hippocampal CSD at different lags from inspiration ($N = 10$ mice). Stars: middle molecular layer sink, circles: outer molecular layer sink. **f**, **h** Recording configuration schematic and example respiration and BLA (**f**) or NAc (**h**) LFP traces. **g**, **i** Example frequency-resolved comodulation of respiration and BLA (**g**) or NAc (**i**) LFP power. **j** Respiration phase histogram and spike waveform for example NAc neuron. **k** Cumulative distribution of logZ for all BLA (blue) or NAc (red) cells (BLA: $n = 101$ cells, NAc: $n = 233$ cells). Inset, mean preferred phase distribution for all significantly modulated cells. **l**, **o** Schematic of recording configuration for sensory thalamus (**l**) and V1 (**o**). **m** Phase histograms and spike waveforms for example midline and sensory thalamus units. **n** Cumulative distribution of the modulation strength for all thalamic neurons ($n = 802$ cells). Inset, preferred phase distribution for all significantly phase-modulated sensory and midline thalamic cells. **p** Spectral V1 LFP power and coherence with respiration. **q** Cumulative distribution of the modulation strength for all V1 PNs and INs. Inset, preferred respiration phase distribution for all significantly phase-modulated neurons. **r** Translaminar phase modulation magnitude and cell percentage. **s** Example inspiration-triggered V1 CSD, compared with visual stimulation-triggered CSD. Inset, the spatial distance between L4 sinks ($N = 5$ mice). Stars indicate significant phase modulation levels (*$P < 0.05$; **$P < 0.01$; ***$P < 0.001$). Shaded areas, mean ± SEM. For box plots, lines correspond to the median, bottom, and top quartile, and whiskers to extremes minus quartile.

fine-timescale modulation of downstream regions by respiration-entrained local and/or OB-originated (at least in mPFC) gamma oscillations.

**Efferent and reafferent mechanisms of respiratory entrainment.** These results suggest a mechanistic picture in which the OB reafferent gamma and respiration-locked slow timescale currents are responsible for the observed breathing-associated LFP patterns in the mPFC, consistent with disruption of these LFP patterns after OB lesion or tracheotomy[31,33,63,64]. However, the distributed and massive modulatory effect that breathing had

on unit activity across wide brain regions is at odds with the anatomically-specific synaptic pathways that we identified as responsible for slow and fast currents.

To causally test whether OB reafferent input is the sole origin of the LFP patterns and unit entrainment, we employed a pharmacological approach, that enables selective removal of the reafferent input. A well-characterized effect of systemic methimazole injection is the ablation of the olfactory epithelium (Supplementary Fig. 7a) that hosts the olfactory sensory neurons[65], that respond to both odors and mechanical stimuli[43]. Effectively, this deprives the OB of the olfactory and respiratory

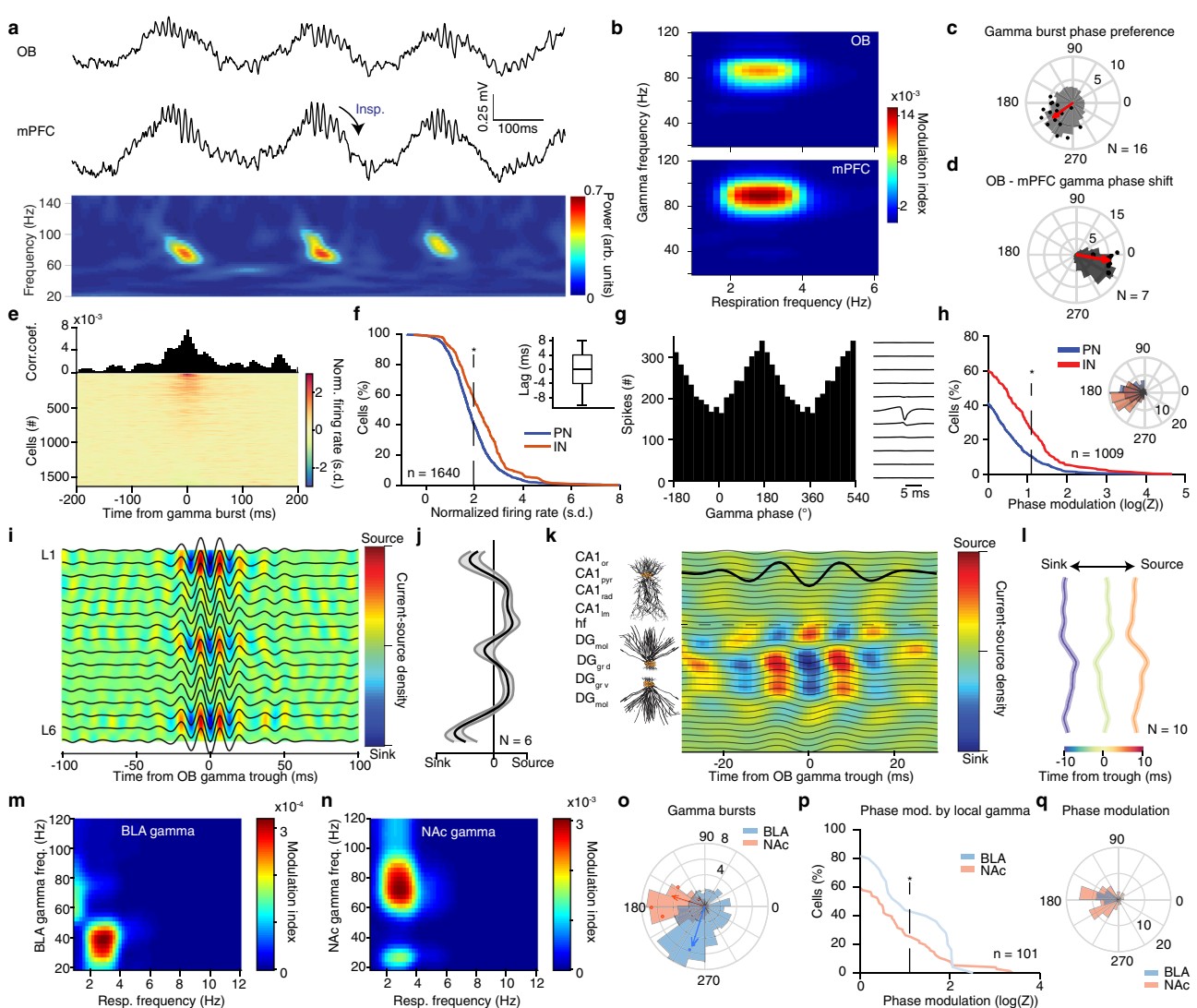

**Fig. 4 Reafferent gamma entrainment of limbic circuits. a** Example LFP traces from OB and mPFC LFP and spectral decomposition of mPFC LFP in the gamma range. **b** Example color-coded modulation strength of OB and mPFC gamma power by respiration phase. **c** Example respiratory phase distribution of mPFC gamma bursts (gray) and average (red arrow) preferred phase and modulation strength (logZ) ($N = 16$ mice). **d** Example phase-shift distribution between OB and mPFC gamma filtered traces (gray) and average phase shift and phase-coupling strength (logZ, red dots) ($N = 7$ mice). **e** Example and color-coded gamma burst triggered histograms for all mPFC cells ($n = 1640$ cells). **f** Cumulative distribution of gamma-triggered firing of mPFC PNs ($n = 1250$ cells) and INs ($n = 390$ cells). Inset, time-lag between gamma burst and peak firing probability for all significantly responsive cells. **g** Gamma phase histogram and spike waveform for one example unit. **h** Cumulative distribution of modulation strength (logZ) by the local mPFC gamma phase for all PNs (blue, $n = 685$ neurons) and INs (red, $n = 324$ neurons). Inset, mean preferred phase distribution of all significantly modulated PN and IN cells. **i, j** Example (**i**) and average zero-lag (**j**) OB gamma-triggered translaminar mPFC CSD. ($N = 6$ mice). **k, l** Example and average OB-gamma-triggered hippocampal CSD at different time-lags from OB gamma trough ($N = 10$ mice). Horizontal line: fissure. **m, n** Example phase-power modulation of BLA (**m**) and NAc (**n**) gamma by respiration. **o** Example distribution of respiration phase for BLA and NAc gamma bursts (histogram) and mean preferred phase of gamma occurrence and modulation strength (dots; BLA, blue, $N = 3$ mice; NAc, red, $N = 4$ mice). **p** Cumulative distribution of modulation strength by local gamma phase of spikes for all BLA (blue, $n = 25$ cells) and NAc cells (red, $n = 76$ cells). **q** Distribution of the mean preferred gamma phase for each significantly modulated BLA and NAc cell. Star indicates significance (*$P < 0.05$). Shaded areas, mean ± SEM. For box plots, lines correspond to the median, bottom, and top quartile, and whiskers to extremes minus quartile.

inputs, while leaving the breathing rhythm generators (Supplementary Fig. 7b–e), as well as the bulbar circuits intact, enabling us to study the contribution of reafferent input on the brain activity in freely-behaving mice. This manipulation in both freely-behaving and head-fixed mice eliminated the respiration-coherent and spectrally-narrow prefrontal slow oscillatory LFP component (Fig. 5a–d and Supplementary Fig. 7g, h), consistent with the disappearance of the CSD sink in deep layers (Fig. 5r), while at the same time significantly reduced the power and correlation of prefrontal gamma oscillations to those in the

olfactory bulb and abolished their entrainment by the breathing rhythm (Fig. 5f, g and Supplementary Fig. 6h), without altering the respiratory dynamics (Supplementary Fig. 7b–r). These results confirm the hypothesis that a respiratory olfactory reafference (ROR) is responsible for the rhythmic cortical LFP during offline states[31,33,63,64,66,67].

Surprisingly, the olfactory deafferentation (OD) left most prefrontal and thalamic neurons modulated by breathing, although the strength of modulation was reduced (Fig. 5h, i), suggesting that a so-far undescribed and yet-to-be-determined

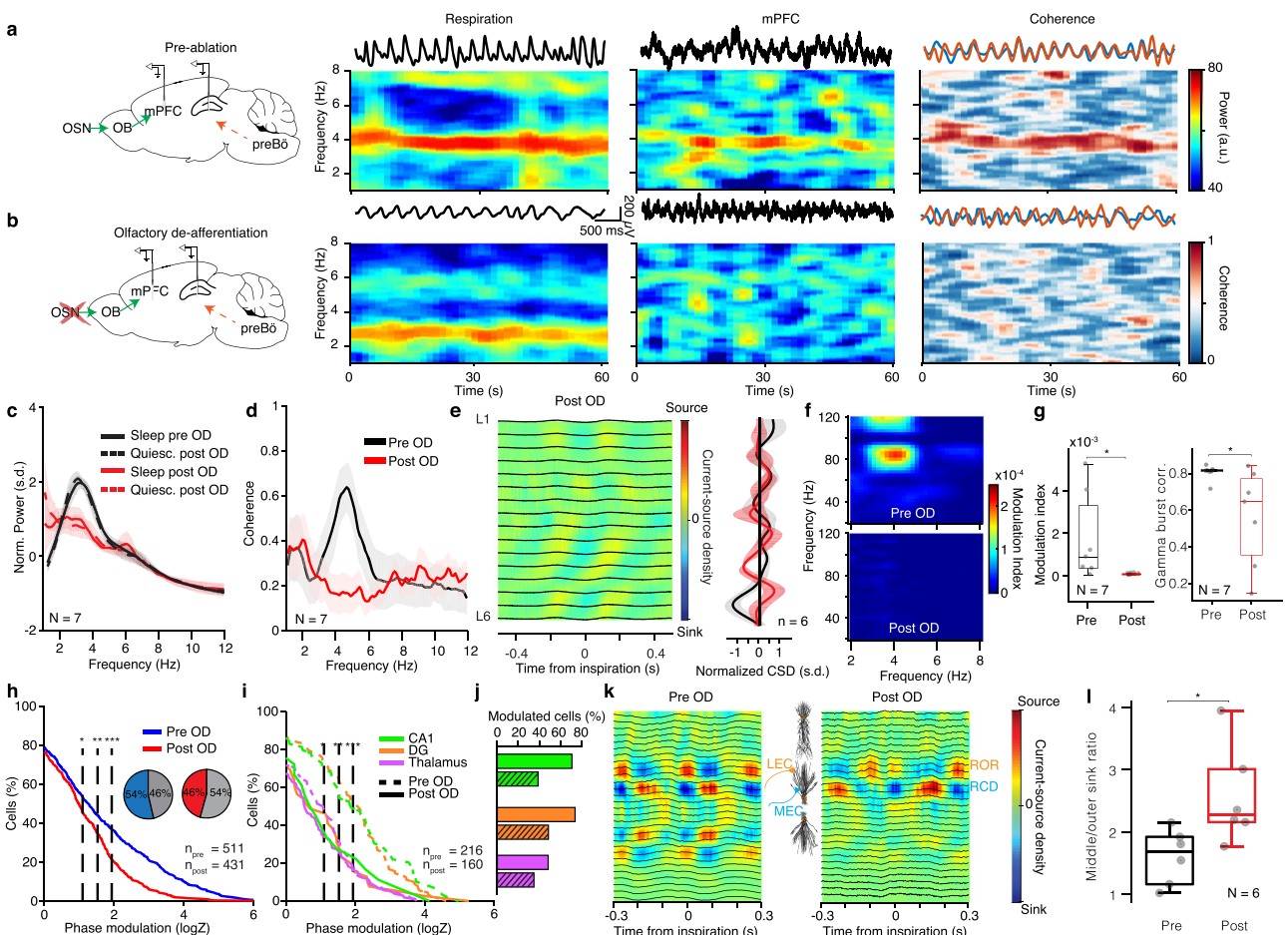

**Fig. 5 Reafferent respiratory input underlies respiratory LFP entrainment. a**, **b** Left, schematic of the manipulation strategy. Right, for example time-frequency decomposition of power and coherence between respiratory and mPFC LFP signals during quiescence before (**a**) and after (**b**) OD. **c** Average normalized mPFC power spectra before and after OD ($N = 7$ head-fixed mice). **d** Coherence spectrum between respiration and mPFC LFP before and after OD ($N = 7$ mice). **e** Left, example inspiration-triggered CSD of the mPFC LFP during quiescence and sleep after OD. Right, average normalized CSD at zero-lag ($N = 6$ mice). **f** Example power-phase modulation of mPFC gamma oscillations before (top) and after (bottom) OD. **g** Average mPFC power-phase modulation strength of ~80 Hz gamma oscillations (left) and gamma burst cross-correlation between mPFC and OB bursts (right) ($N = 7$ mice; Wilcoxon signed-rank two-sided test: before vs. after OD, $P = 0.0312$). **h** Cumulative distribution of modulation strength for all mPFC neurons pre and post-OD (Pre: $n = 511$ cells; Post: $n = 431$ cells). Inset, percentage of significantly phase-modulated cells before and after OD. **i** Cumulative distribution of modulation strength for CA1, DG, and somatic thalamus neurons before and after OD. **j** Percentage of significantly phase-modulated cells before and after OD. **k** Example inspiration-triggered CSD of the dorsal hippocampus LFP before (left) and after (right) OD. **l** Average ratio of the middle to outer molecular layer sink depth ($N = 6$ head-fixed mice; Wilcoxon two-sided signed-rank test: before vs. after OD, $P = 0.0312$). Shaded areas, mean ± SEM arb. units arbitrary units, s.d. standard deviations, n.s. not significant. Shaded areas, mean ± SEM. Stars indicate significance levels (*$P < 0.05$; **$P < 0.01$; ***$P < 0.001$). s.d. standard deviations, arb. units arbitrary units, OD olfactory deafferentation. For box plots, the middle, bottom, and top lines correspond to the median, bottom, and top quartile, and whiskers to lower and upper extremes minus bottom quartile and top quartile, respectively. Source data are provided as a Source Data file.

anatomical-physiological mechanism of centrifugal efference copy provided by brainstem circuits (termed ascending respiratory corollary discharge (RCD)) is responsible for the massive entrainment of limbic neurons. Interestingly, following OD, freely-behaving mice exhibited intact memory and fear expression, suggesting that freezing behavior does not rely on the ROR input (Supplementary Fig. 7i–k). Dorsal hippocampal neurons in head-fixed mice were somewhat stronger affected by the ablation, yet >40% of cells were still significantly phase modulated by breathing (Fig. 5i, j), indicating a differential degree of contribution of RCD and ROR to unit firing across mPFC, hippocampus, and thalamus. In contrast to the prefrontal CSD, the olfactory deafferentation led to a strong reduction of the outer molecular layer current sink originating in LEC LII in the respiration-locked CSD (Fig. 5k, l), while leaving the current sink

originating in MEC LII and other non-respiration-related CSD patterns intact (Supplementary Fig. 7h), suggesting that the LEC input is driven by ROR, while MEC input is driven by RCD.

**Hippocampal network dynamics are modulated by breathing.** From the results so far, it is clear that hippocampal neuronal activity during offline states is massively modulated by breathing, in part via afferent entorhinal inputs to the DG. However, during quiescence and slow-wave sleep, hippocampal activity is characterized by recurring nonlinear population events such as sharp-wave ripple (SWR) complexes and dentate spikes (DS), that organize the local spatiotemporal population activity implicated in the replay of memories[18] and known to be tightly temporally correlated with slow cortical oscillations during sleep[7].

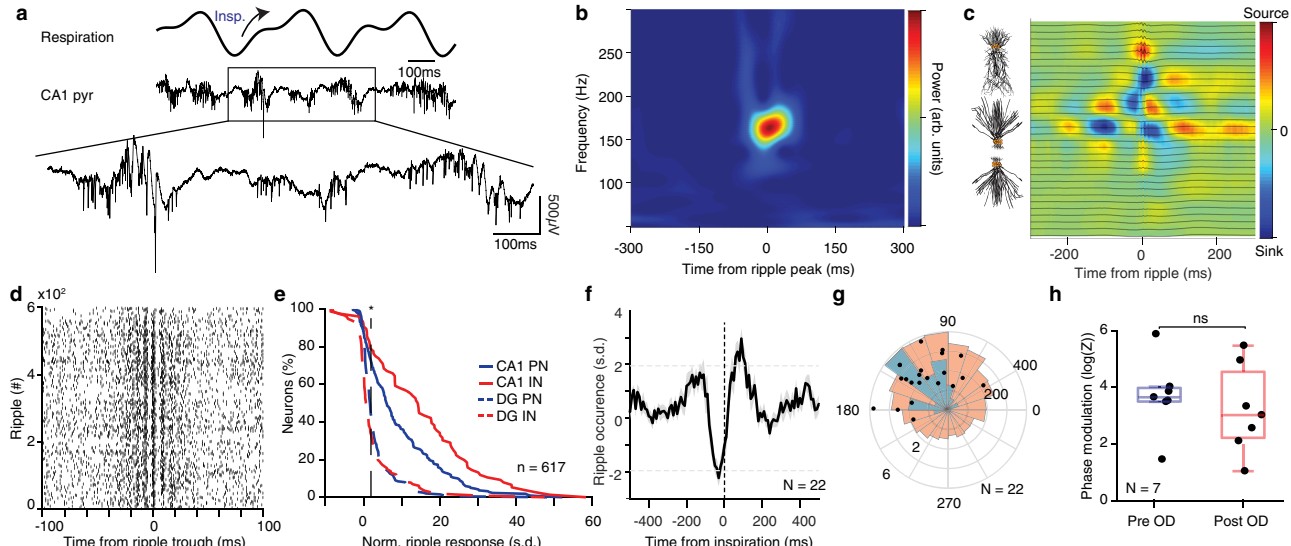

**Fig. 6 Breathing modulates hippocampal ripples. a** Example traces of the respiratory signal and CA1 pyramidal layer LFP. In the magnified LFP signal, ripple events and the associated spiking activity can be observed. **b** Average ripple-triggered time-frequency wavelet spectrogram of the CA1 pyramidal layer LFP from one example animal. **c** Schematic of the CA1 pyramidal and granular cells somatodendritic domains aligned to the average ripple-triggered CSD profile of the hippocampal LFP activity for one example animal. **d** Ripple-triggered spike train of an example dCA1 neuron exhibiting evoked response and a characteristic oscillatory firing pattern. **e** Cumulative distribution of the ripple-triggered normalized firing of CA1 and DG PNs (CA1, $n = 220$ cells; DG, $n = 202$ cells) and INs (CA1, $n = 76$ cells; DG, $n = 119$ cells). **f** Average cross-correlation between inspiratory events and ripple occurrence ($N = 22$ freely-behaving and head-fixed mice). Dashed horizontal lines indicate the significance levels. **g** Distribution of the respiratory phase of occurrence of individual ripple events for one example animal (red; $n = 4813$ ripples) and preferred phase distribution for the population (blue; $N = 22$ sessions). Overlaid, distribution of average phase and modulation strength for ripples (black dots). **h** Phase modulation of ripples before and after OD ($N = 7$ mice). s.d. standard deviations, arb. units arbitrary units, n.s. not significant, OD olfactory deafferentation. Shaded areas, mean ± SEM. Star indicates significance level (*$P < 0.05$). Source data are provided as a Source Data file.

During CA1 ripples (Fig. 6a–c and Supplementary Fig. 8a), CA1 PN and IN, and to a lesser extent DG cells, were strongly (Fig. 6e) and rhythmically (Fig. 6d) activated[10]. Ripple occurrence during quiescence and sleep in head-fixed mice was strongly modulated by breathing[68], biased towards the post-inspiratory phase, while ripples were suppressed during exhalation (Fig. 6f, g). Following olfactory deafferentation, ripples remained significantly phase modulated by breathing (Fig. 6h), suggesting that RCD is the main source of their modulation. Keeping up with the role of the entorhinal input in mediating respiratory drive on ripples, we observed a consistent relationship between the magnitude of the current sink in DG mol. layer directly preceding ripple occurrence and the phase within the respiratory cycle of the ripple occurrence (Supplementary Fig. 8c, d), suggesting that ripples occurring at the preferred phase of respiration follow a strong MEC LII input.

Another prominent hippocampal offline state-associated pattern are dentate spikes, defined as large positive potentials in the DG hilus region[52], which are believed to occur in response to strong inputs in the molecular layer, such as during entorhinal UP states (Supplementary Fig. 5e, f)[12,69]. During DS, both DG (~70% PN and IN), CA1 (~40% PN and ~70% IN), and mPFC (~40% PN and ~70% IN) cells were strongly excited (Supplementary Fig. 5g). Consistent with respiratory entrainment of the entorhinal inputs, the occurrence of DS was strongly modulated by the breathing phase, both before and after OD, with the majority of events occurring after inspiration (Supplementary Fig. 5h–j).

**Breathing organizes prefrontal UP states and hippocampal output.** Similar to the hippocampus, during quiescence and slow-wave sleep neocortical circuits exhibit nonlinear bistable dynamics in the form of DOWN and UP states. We posited that

the strong rate of modulation of prefrontal neural activity by breathing would bias these dynamics. To test this prediction, we identified prefrontal UP and DOWN states during quiescence and sleep in head-fixed mice based on the large-scale population activity and characterized their relationship with the phase of the ongoing breathing rhythm (Fig. 7a). Both UP and DOWN state onsets were strongly modulated by the breathing phase and time from inspiration (Fig. 7b–f), while the magnitude of modulated UP states followed the profile of UP state onset probability (Fig. 7c). In line with the results on ripples and prefrontal units, UP and DOWN state modulation was not affected by olfactory deafferentation, suggesting that RCD is sufficient to organize the cortical dynamics (Fig. 7e, f).

Previous observations during sleep in rats identified a temporal correlation between ripple occurrence and cortical DOWN/UP state complexes[7,70,71], however, the mechanism underlying this correlation remains unknown. We found that ripples during sleep in head-fixed mice preceded the termination of prefrontal UP states and onset of the DOWN states both before (Fig. 7g) and after deafferentation (Fig. 7i), with ripples associated with UP state termination and immediately preceding a DOWN state tended to occur in the early post-inspiratory phase (Fig. 7h). This is consistent with the respiration-related synaptic input to the DG middle molecular layer preceding ripple events (Fig. 6c), which persists following OD (Supplementary Fig. 8b), which suggests RCD-mediated coordination of SWR occurrence with the cortical UP states known to be mediated via the MEC[12] (Supplementary Fig. 8c, d).

Ripple output is known to recruit prefrontal neural activity[70,71]. In agreement with this, hippocampal ripples during quiescence and sleep in head-fixed mice evoked a response in prefrontal LFP and gave rise to an efferent copy detected as a local increase in fast oscillatory power in the PFC LFP (Fig. 8a)[72]. In

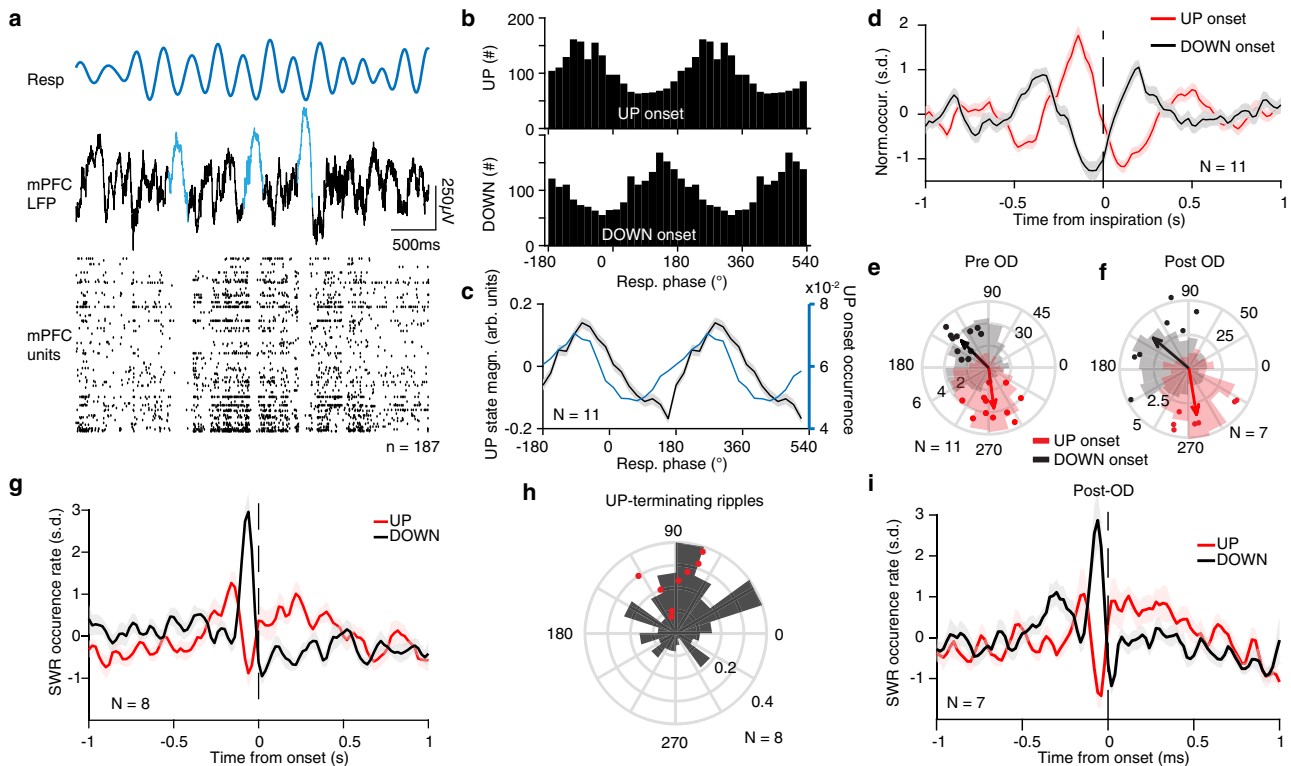

**Fig. 7 Breathing modulates cortical UP/DOWN state transition. a** Example traces of respiration signal (top), mPFC LFP (middle trace), and spike trains of 187 simultaneously recorded mPFC units during sleep. Three typical delta waves and the corresponding DOWN states of the neuronal population are marked with blue. **b** Example distributions of the breathing phase of UP (top) and DOWN (bottom) state onsets. **c** Normalized power (black) and occurrence rate (blue) for prefrontal UP states ($N = 11$ freely-behaving and head-fixed mice). **d** Cross-correlation of UP and DOWN state onsets with respect to inspiration. **e**, **f** Distribution of preferred breathing phase and phase modulation magnitude (logZ) of UP and DOWN states before (**e**) and after (**f**) OD. (pre: $N = 11$ mice, post: $N = 7$ mice). Histograms are for one example session (scale is a count of UP/DOWN states). **g** Normalized occurrence rate of SWR as a function of time from UP or DOWN state onset ($N = 8$ mice). **h** Example (black) and distribution of preferred breathing phase of SWR occurrence, for SWRs that are terminating a UP state (red dots, $N = 8$ mice). **i** Normalized occurrence rate of SWR as a function of time from UP or DOWN state onset after OD ($N = 7$ mice). Note that the observed pattern is identical to the pre-OD shown in (**g**). For box plots, the middle, bottom, and top lines correspond to the median, bottom, and top quartile, and whiskers to lower and upper extremes minus bottom quartile and top quartile, respectively. s.d. standard deviations, arb. units arbitrary units, OD olfactory deafferentation. Shaded areas, mean ± SEM. Star indicates significance level (*$P < 0.05$).

response to ripple events, ~14% of prefrontal PNs and ~42% of INs exhibited increased firing (Fig. 8b, d, e and Supplementary Fig. 8l, o), while a small fraction of these cells was also rhythmically modulated by the ripple phase (Fig. 8c). In parallel, ~69% of NAc cells were significantly driven by ripple events (Fig. 8h, i and Supplementary Fig. 8o), while in both mPFC and NAc there was a great overlap between cells that were phase modulated by breathing and those that were responsive to ripples (Fig. 8j and Supplementary Fig. 8o). Interestingly, the phase of breathing modulated the excitability of both mPFC and NAc, as revealed by the modulation of ripple-evoked activity magnitude by the phase of breathing (Fig. 8e, i), as well as the fine-timescale (10 ms) co-firing between CA1 and mPFC (Fig. 8f) and within prefrontal regions (Supplementary Fig. 8e).

Given the observation that prefrontal population activity is limited by the respiratory modulation on a low-dimensional manifold (Fig. 2d), we investigated the effect of ripples occurring during inspiration on the trajectory of the neural population activity. Interestingly, ripples transiently perturbed cortical dynamics, which quickly returned to the respiration-driven limit cycle (Fig. 8g).

Given the mutual connectivity between the cortical networks and the hippocampus, delineating the causal role of the joint respiratory modulation for the coordination of SWR and UP/DOWN dynamics only by passive observation of the tripartite

correlation of cortical, hippocampal, and respiratory dynamics is difficult. We thus opted for optogenetically generating ripple oscillations, an experimental manipulation that enabled us to decorrelate the timing of hippocampal ripples from the breathing cycle and thus allowed us to observe the effect of respiratory modulation of prefrontal circuits on this coordination. To achieve that, we expressed excitatory opsin (AAV2/9-CaMKIIa-ChE-TA(E123T/H134R)-eYFP-WPRE.hGH) in the dCA1 (Fig. 9b) and delivered low-intensity half-sine wave light stimulation in head-fixed mice (Fig. 9a, c). Light stimulation resulted in the rhythmic depolarization of dCA1 neurons (Fig. 9d and Supplementary Fig. 8g) and the generation of short-duration, high-frequency oscillations (termed opto-ripples) (Fig. 9c and Supplementary Fig. 8i)[56,73] with peak frequency ~75 Hz (Fig. 9e and Supplementary Fig. 8g, h). These oscillations induced evoked activity in the mPFC qualitatively similar to the one described during intrinsic ripples, both at the LFP level, that was consistent with opto-ripple-evoked K-complex (Fig. 9f and Supplementary Fig. 8j), unit level (Fig. 9h and Supplementary Fig. 8k), and population dynamics (Fig. 9g). Interestingly, the magnitude of the prefrontal depolarization in response to opto-ripples was modulated by the ongoing phase of the breathing cycle when opto-ripple was generated, similar to intrinsic ripples (Fig. 9i, Supplementary Fig. 8i). The persistence of the excitability modulation of mPFC following OD (Fig. 9i), further reinforces

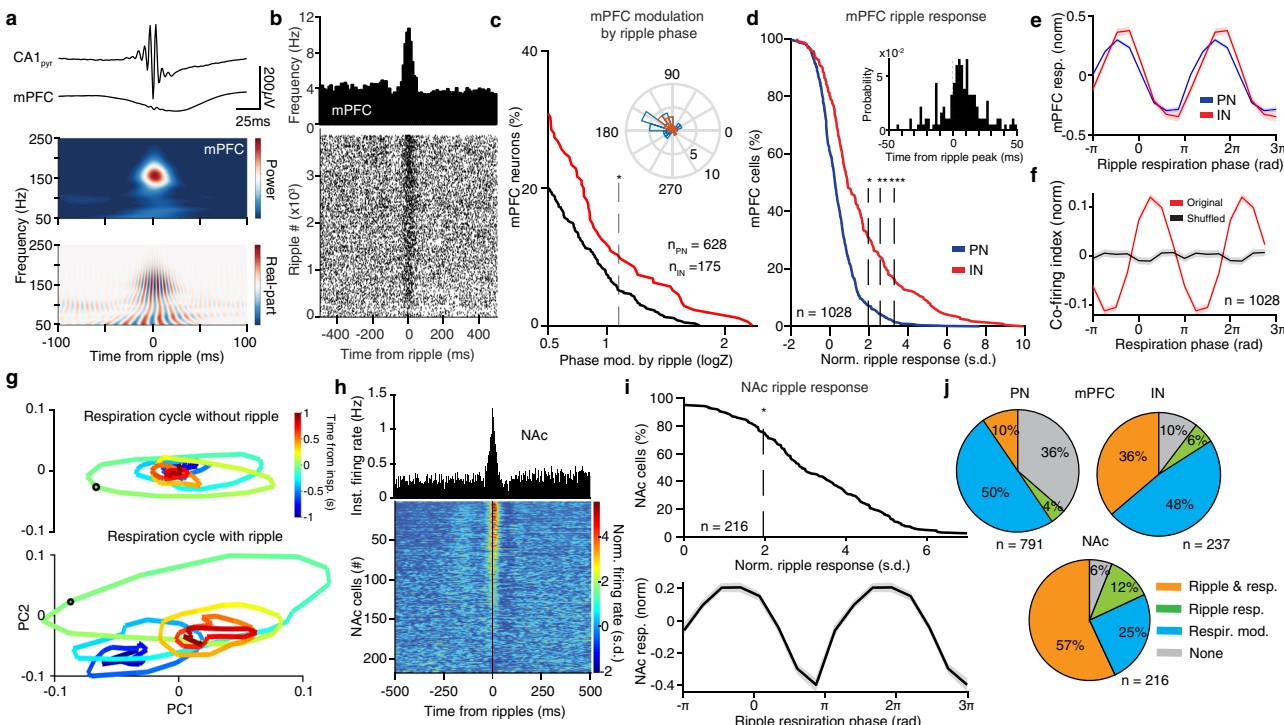

**Fig. 8 Breathing modulates the cortico–hippocampal interaction. a** Example is ripple-triggered $CA1_{pyr}$ and mPFC LFP traces and wavelet spectral decomposition of mPFC LFP power (upper) and real part (lower) ($n = 3162$ ripples). **b** Example mPFC unit spiking across ripples (bottom) and cross-correlogram of unit firing to ripple occurrence (top). **c** Cumulative distribution of the modulation strength by ripple oscillation phase for all mPFC PNs ($n = 628$ cells) and INs ($n = 175$ cells). Inset, preferred phase distribution for all significantly phase-modulated cells. **d** Cumulative distribution of the ripple-triggered firing of mPFC PNs ($n = 791$ cells) and INs ($n = 237$ cells) in response to ripples ($N = 11$ mice). Inset, distribution of relative time of peak firing. **e** Prefrontal ripple-evoked response for PN (blue traces) and IN (red traces), as a function of the breathing phase of occurrence of ripple. **f** Co-firing index for all pairs of dCA1 and mPFC cells ($n = 14,412$, $N = 13$ mice) compared to shuffle control. **g** Example trajectory of the mPFC population dynamics around inspiration without (top) and with (bottom) a ripple oscillation occurring during that period in the oscillation. Black circle denotes time 0, pseudocolor codes for the time from inspiration onset. **h** Cross-correlation of firing with respect to ripple time for one example NAc unit (top) and color-coded cross-correlograms for all NAc cells ($n = 216$ cells). **i** Top, cumulative distribution of the ripple-triggered normalized firing of NAc cells ($n = 216$ cells) in response to ripples ($N = 4$ mice). Bottom, ripple-evoked response as a function of the breathing phase of occurrence of ripple. **j** Top, pie charts indicating the percentage of all mPFC PNs (left; $n = 791$ cells) and INs (right; $n = 237$ cells) that are either phase modulated by respiration (resp. mod), responding significantly to ripples (ripple resp.), being both significantly modulated by breathing and significantly responsive to ripples or neither. Bottom, similarly, for all NAc cells ($n = 216$ cells). s.d. standard deviations, arb. units arbitrary units. Shaded areas, mean ± SEM. Stars indicate significant phase modulation levels (*$P < 0.05$; **$P < 0.01$; ***$P < 0.001$).

the notion that RCD is the main mechanism behind this phenomenon. The smaller amplitude of the opto-ripple-evoked response is likely due to the spatially-defined optogenetic stimulation. A qualitatively similar, though with distinct breathing phase sensitivity, the result can be observed in the visually-evoked responses of the visual cortex in head-fixed mice (Supplementary Fig. 8g, h), which constitutes a distinct, LGN-mediated, way of externally probing ongoing cortical excitability.

These results suggest that breathing rhythmically modulates cortical and hippocampal excitability and biases network dynamics in these circuits, as well as their interactions. The joint modulation of these dynamics results in the coordination of emitter and receiver circuits and might contribute to the mechanisms of control of information flow that enables the integration and segregation of information flow and processing across the network.

## Discussion

The propagation of information across distinct neuronal networks is facilitated by the coordination of these dynamics between brain regions[5,7]. In this study, we demonstrate that the respiratory rhythm, via a centrifugal corollary discharge, acts as a functional oscillatory scaffold and provides a unifying global

temporal coordination of neuronal firing and network dynamics across cortical and subcortical networks during offline states. The comprehensive phase-resolved picture of this synchronization provides the basis for mechanistic theories of information flow across the limbic system (Fig. 10b).

**Respiratory corollary discharge couples global brain circuits.** Using pharmacological manipulations paired with large-scale recordings, we characterize a joint mechanism of respiratory entrainment, consisting of an efference copy of the brainstem respiratory rhythm or vagal re-afferents (respiratory corollary discharge; RCD) that underlies the neuronal modulation of brain regions and a respiratory olfactory reafference (ROR) that contributes to the modulation and accounts for respiration-locked LFP signals (Fig. 10a). We have demonstrated the global entrainment of limbic neuronal activity by the respiratory rhythm that is mediated by the intracerebral RCD, likely originating in the brainstem rhythm generator circuits and being unaffected by olfactory deafferentation.

Brainstem circuits are well-known to generate breathing rhythm[74,75], and at the same time send massive diffuse ascending projections to the forebrain[76,77], thus the most parsimonious explanation of the observed phenomenon is a directional drive

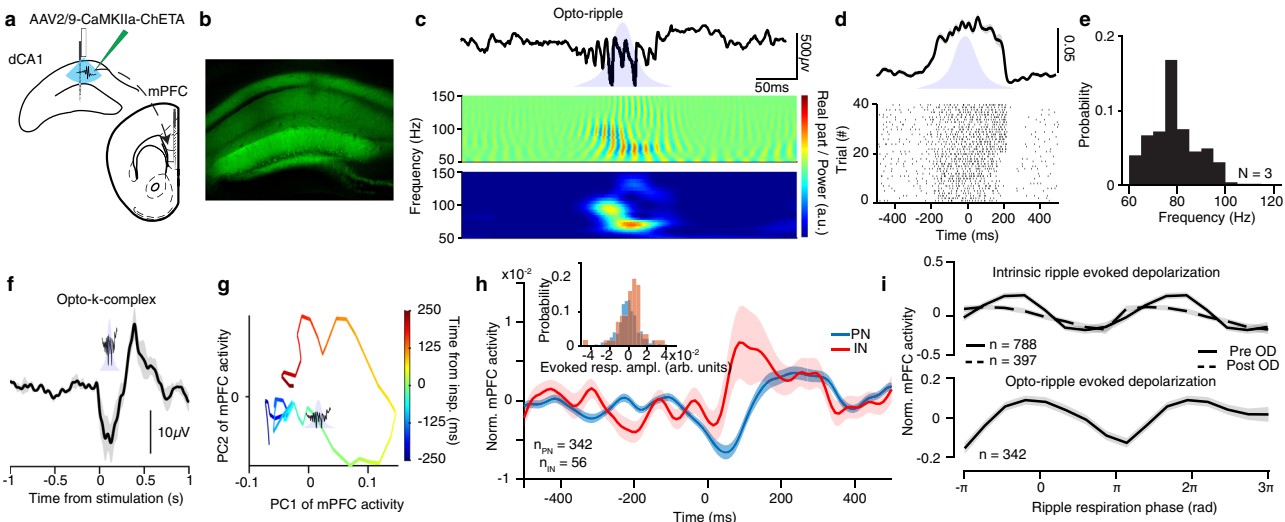

**Fig. 9 Respiratory modulation of cortical excitability. a** Schematic of the experimental design for testing the effect of optogenetically induced hippocampal ripples on the prefrontal network. An excitatory opsin (AAV2/9-CaMKIIa-ChETA) was injected in the dCA1 and a silicon probe paired with an optic fiber was inserted to generate opto-rippled and to record the induced activity. Simultaneously, a silicon probe was implanted in the mPFC to record the effect of the intrinsic and optogenetically induced ripples. **b** Example histological reconstruction of the opsin expression in the dorsal hippocampus. **c** Example trace (top) and wavelet real part (middle) and power (bottom) of the oscillation generated in the dCA1 pyramidal layer during the optogenetic stimulation. Note the similarity with ripple oscillations. **d** Cross-correlation of firing with respect to optogenetic stimulation for one example dCA1 neuron and raster-plot for all events. **e** Probability distribution of the frequency of evoked oscillations during light stimulation (N = 3 mice). **f** Example average evoked response in the mPFC LFP triggered on the opto-ripple generation in the dCA1. **g** Example trajectory of the mPFC population around the time of inspiration when an opto-ripple is generated during inspiration. **h** Evoked response in the mPFC neuronal population triggered on the opto-ripple generation in the dCA1. Inset, distribution of the evoked response amplitude for all cells (n = 398 cells). **i** Z-score normalized prefrontal neuron evoked activity magnitude (n = 788/397 cells) in response to intrinsic for pre and post-OD conditions (top) and optogenetically-generated (n = 342 cells) (bottom) ripples, as a function of the respiratory phase of occurrence of the ripple. Shaded areas, mean ± SEM. arb. units arbitrary units.

from the brainstem to the forebrain. While descending feedback inputs[78], might modulate breathing and contribute to the phenomenon, they are not serving the rhythm generation function, which is known to be implemented in the brainstem circuits that provide a causal effect on the forebrain circuits[74]. OD experiments result in a virtually absent spectral peak in the prefrontal LFP and reduction of gamma and unit entrainment across different circuits (Fig. 5), while respiratory activity is unchanged. This strongly suggests the same directionality of the phenomenon as suggested by the analytical methods.

These results highlight the global extent and significance of this modulation, given the crucial role that the interaction between limbic structures plays for emotional processing and memory functions. Although the pathways mediating RCD remain unknown, we speculate that ascending long-range somatostatin-expressing cells, projecting from the preBötzinger complex to the thalamus, hypothalamus, and basal forebrain[77] or the locus coeruleus[76], are probable pathways for this widespread modulation. An alternative source of such modulation could be ascending vagal re-afferents[79,80]. A disinhibition-mediated mechanism of RCD would be consistent with the lack of prominent LFP sources in the absence of ROR following olfactory deafferentation, similar to the mechanism of disinhibitory pacing by the medial septum of the entorhinal-hippocampal system during theta oscillations[8]. The global and powerful nature of the RCD suggests that potentially multiple parallel ascending pathways from the brainstem mediate this signal redundantly and calls for future tracing and activity-dependent labeling studies to identify its anatomical substrate.

We suggest that centrifugal modulation by breathing is analogous to the predictive signaling employed in a wide range of neural circuits[81], such as those underlying sensory-motor coordination[82] and likely extends to other brain structures and

brain states. During active behavior, the respiratory phase modulates the processing of olfactory inputs[83,84], while the coupling between breathing and whisking might underlie the coordination of active sampling processes[85], with breathing serving as the reference signal of various orofacial rhythms[86]. The global outreach of RCD to higher-order areas suggests that it might play an important role in the coordination of multi-sensory processing, in sync with orofacial motor output during both passive and active orofacial sampling, thus providing a centrifugal component synchronized with reafferent sensory inputs and respiratory efference copies to orofacial motor centers[85].

**Respiratory entrainment—a potential substrate of fear behavior.** Extending the generality of respiratory rhythm entrainment, we show that fear-related prefrontal 4 Hz oscillations[30,87] are a state-specific expression of this entrainment and originate from the reafferent respiratory entrainment of olfactory sensory neurons by passive airflow[43] (Supplementary Fig. 2).

Fear memory and expression relies on the coordinated activity in the mPFC and BLA. The joint respiratory modulation of prefrontal and amygdala circuits might contribute to this coordination during fear behavior. The pronounced LFP amplitude and narrow respiratory and LFP frequency during this state (Supplementary Fig. 2) is potentially fine-tuning the temporal coincidence of ensemble activity in these regions, resulting in increased coherence and co-firing[30].

Importantly, although prefrontal 4 Hz LFP oscillations originate in fear-associated enhanced breathing, the ROR is not necessary for the expression of innate or conditioned fear behavior, in agreement with a recent report[31]. This suggests the independence of fear expression from the respiratory entrainment or the potential sufficiency of RCD for the expression of fear behavior.

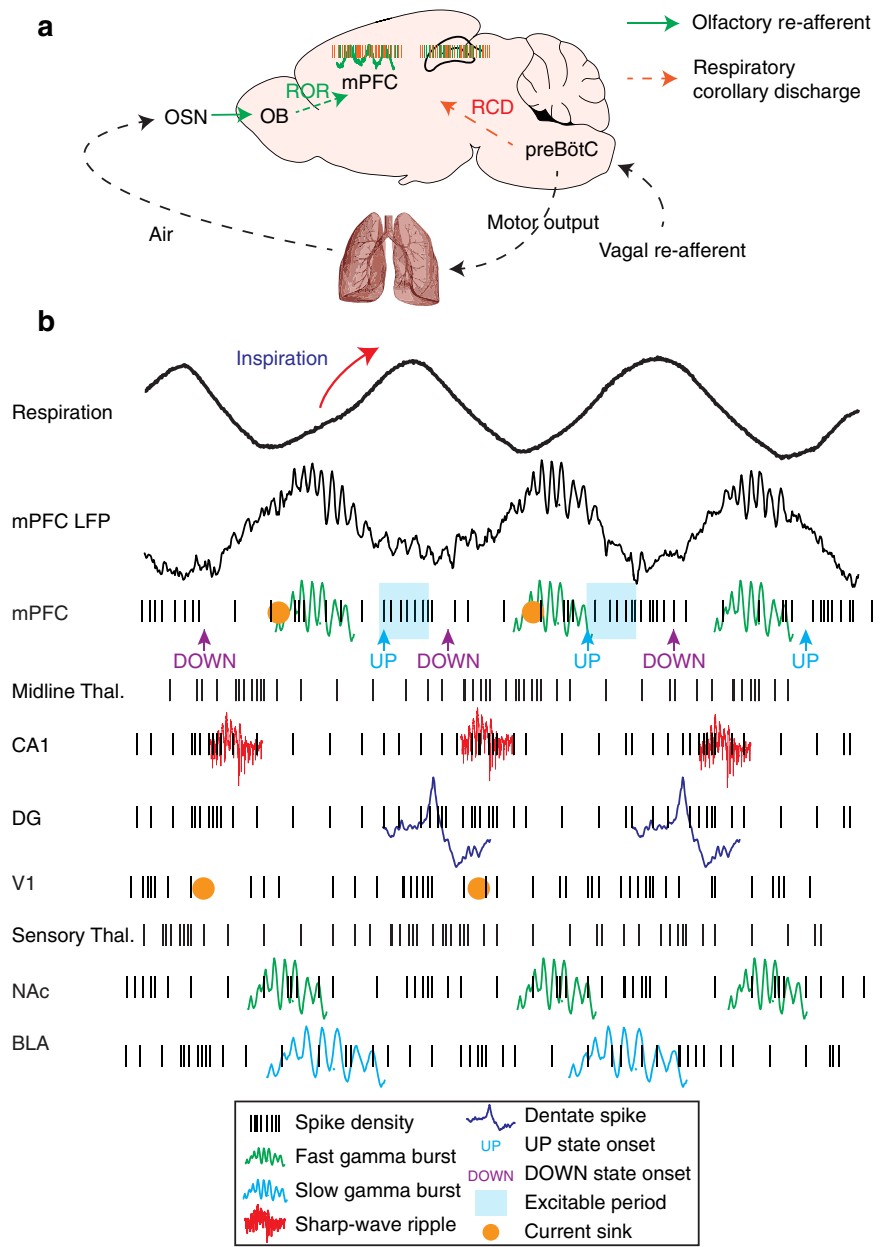

**Fig. 10 Breathing organizes network dynamics across limbic structures. a** Schematic depiction of the efferent copy pathway carrying the respiratory corollary discharge (RCD) signal and the reafferent pathway carrying the respiratory olfactory reafferent (ROR) signal. **b** Summary schematic of the network dynamics organized by breathing throughout all structures studied. Black traces: LFPs; Black ticks: neuronal spikes; Green traces: Fast (~80 Hz) gamma; Cyan traces: Slow gamma (~40 Hz); Red traces: CA1 Ripples; Blue traces: Dentate spikes; Orange dots: CSD sinks (mPFC deep layers, V1 LGN input, and DG middle molecular layer); Blue shadow: Cortical excitability period.

Consistent with such mechanism, the optogenetic induction of 4 Hz oscillations in prefrontal circuits is sufficient to drive fear behavior in naïve animal[30], raising the possibility that this effect is mediated by the bidirectional interaction of prefrontal networks with the respiratory centers via top-down projections to periaqueductal gray and the ascending feedback via RCD giving rise to system-level resonance at breathing frequency. This sets the stage for future investigations of the interaction between the RCD and ROR in limbic networks and, in turn, the top-down modulation of breathing and emotional responses.

**Breathing rhythm mediates synchronization of the cortico-hippocampal dynamics - Implications for models of memory consolidation.** Mechanistic theories of systems memory

consolidation posit that synchronous network activity plays a central role in attractor dynamics underlying the reactivation of memory traces and coordination of information flow. The original two-stage model proposed a central role of hippocampal SWR in broadcasting memory traces[45], while the fine-timescale relationship of SWR and neocortical DOWN/UP state dynamics[11,12] led to an update of the model that posited active functional role of slow oscillations in bringing cortico-hippocampal dynamics together and thus providing coordinated windows of opportunity for information flow[5,7]. While a critical role of SWR and their coordination with DOWN/UP states for memory consolidation has been demonstrated[21,23], there is no established mechanism mediating the coordination of SWR with distributed neocortical DOWN/UP state dynamics and, importantly, other brain structures that take part in memory consolidation. Here we demonstrated that the respiratory rhythm coordinates cortico-

hippocampal dynamics as well as unit activity across the limbic system, with RCD being a sufficient mechanism. Furthermore, we show that respiratory coordination modulates the responsiveness of downstream prefrontal and nucleus accumbens populations to hippocampal SWR and CA1-mPFC unit co-firing, suggesting that respiratory phase biases interaction between these circuits. Importantly, opto-ripple-based experiments demonstrate that it is the breathing cycle and not other covariates of endogenous ripples that modulate cortical excitability, while OD experiments suggest that this phenomenon is mediated by RCD. This respiration-coordinated circuit interaction model extends previously observed bivariate correlations between SWR and prefrontal[70,88], accumbens[46], and amygdala[14] neural activity, as well as respiratory entrainment of hippocampal units[32,48,49] and ripples[68], which can, in part, be accounted for by the respiratory comodulation demonstrated here. This does not imply the necessity of respiratory modulation for reactivation per se, or that reactivation in limbic structures occurs specifically in temporal association with SWR, but that temporal correlation between population activity in all the respected structures, in addition to direct connectivity, is comodulated by a common rhythm that can support effective information transfer.

Based on CSD analysis and pharmacological manipulation, we suggest that RCD and ROR inputs, via MEC and LEC respectively, give rise to hippocampal unit modulation, RCD likely responsible for the emergence of dentate spikes and SWR respiratory-entrainment. Depending on its strength, the RCD input, either via feed-forward inhibition of the CA3[89] or forward excitation can, respectively, delay or advance SWRs within the respiratory cycle. In parallel, the ripple-driven recruitment of prefrontal neurons likely triggers the resetting of the ongoing UP states by tilting the bias between excitation and inhibition[90] and results in a feedback re-entrance to the entorhinal-hippocampal network[12]. In concert with this bidirectional interaction, the intrinsic RCD-mediated comodulation of SWR and DOWN/UP state complexes by the breathing phase is likely setting up the functional sequences of these nonlinear dynamics that are critical for memory consolidation. Indeed, breathing rhythm biases the occurrence of SWR prior to DOWN state onset (Fig. 7g)[11,13], while the causal enhancement of this pattern improves memory consolidation[23]. Understanding the causal role of respiratory entrainment in the synchronization of cortico-hippocampal dynamics will require the dissection of circuit mechanisms of RCD and fine-timescale, closed-loop optogenetic perturbations. Our causal optogenetic manipulation of one side of this complex dynamical system identifies an excitability profile of hippocampo–cortical interaction within a breathing cycle and paves the way for future respiration phase-resolved gain of function studies.

While we show here that the respiratory dynamics bias the prefrontal DOWN/UP states via RCD, slow oscillations can emerge in isolated cortical slubs[91] or slices[92] and thus slow oscillations observed at a similar frequency in the thalamocortical system across all mammalian species[28] are not necessarily forced by the breathing rhythm[49,93]. Rather, we suggest considering their interaction from the mechanistic dynamical systems perspective of coupled bistable nonlinear dynamical systems. Indeed, leading models of the generation of neocortical UP states from DOWN states[94] or inspiratory bursts from expiratory silence in preBötzinger circuits[74] suggest that both phenomena rely on regenerative avalanches, due to recurrent connectivity, that are followed by activity-dependent disfacilitation. Given that neocortical slow oscillations can be locally generated[7,95], are globally synchronized by the thalamic input[96] and propagate across the neocortex[12,97], RCD biasing of the cortical DOWN/UP state complex could be considered as an extension of a global system of mutually-coupled nonlinear oscillators. The persistent synchronous output of the respiratory oscillator and its

marginal independence of the descending input might provide a widespread asymmetric bias to both cortical DOWN/UP states and hippocampal SWR across offline states of different depth. It is likely, however, that via descending cortical projections[78], cortical SO provides feedback to the pontine respiratory rhythm-generating centers and thus the interaction between respiratory dynamics and slow oscillations could be bidirectional. Further, transient cessation of breathing during apnea or extended expiration could provide windows of low impact of breathing rhythm on the dynamics, whereas frequency fluctuations of the respiratory rhythm compress or stretch the ongoing cortical dynamics (Supplementary Fig. 3g). The functional implications of these phenomena for the coordination of network dynamics remain to be studied. Nevertheless, the interactions of cortico-hippocampal dynamics and bivariate interactions between different structures based on their direct connections might be sufficient for providing a high degree of correlation that doesn't decorrelate over short periods devoid of breathing pacing.

Finally, the perpetual comodulation of limbic population activity by breathing also suggests a potential framework for memory consolidation processes that do not rely on deep sleep and the associated synchronous cortical DOWN/UP states, that could explain the mechanism and distinctive role of awake replay in memory consolidation[98,99].

**Olfactory bulb reafferent source of LFP signals and gamma oscillations**. Respiratory olfactory reafferent input, in the form of synchronous recruitment of olfactory bulb by sensory input, organizes internal dynamics in OB circuit[39,40,100] and gives rise to entrainment and macroscopically observed respiration-coherent LFP signals in the downstream structures[33,63,101–103]. Our results extend and provide a mechanistic explanation and interpretation of the emergence of respiration-related LFP signals in these regions, dissociating them from the centrifugal RCD-mediated mechanism of entrainment of neural activity. We further report the reafferent OB origin of local gamma dynamics and their modulation by breathing, as well as the relation between local gamma oscillations and neuronal activity in all structures, which illuminates the origin and role of prefrontal[57,103], BLA[58,104], and NAc[105] gamma oscillations.

Anatomically-resolved analysis of prefrontal OB-generated current sources and unit activity suggests that deep layers and mostly ventral regions are the main targets of OB reafference and give rise to observed LFP signals. The reduction of the dorsoventral LFP profile following OD is consistent with olfactory-related sources of this gradient, rather than brain-motion-related activity due to airflow through the nasal cavity. Although the interpretation of these gradients is challenging given the existence of volume conduction from the OB, these findings suggest a potential functional role of the differential modulation of orbital, prefrontal, and cingulate regions and is worthy of future investigation. Importantly, the propagation of respiration-driven excitation to distant structures is jointly driven by both ROR and RCD, the former giving rise to observed LFP signals that disappear after OD, as shown here, or OB ablation[31,33,63]. Thus, due to the fast breathing frequency in mice, slow and/or delta power cortical LFP signals are in part, and depending on the synaptic distance from the OB, contaminated by ROR, making the direct analysis of slow oscillation or delta waves based on LFP signal alone unreliable. Further investigations are required to dissociate the differential functional role of ROR and RCD in recruiting brain circuits to the breathing rhythm.

This joint modulation of downstream circuits suggests a model in which synchronous ROR inputs reach the target regions in sync with RCD-coordinated local activity. These oscillations

potentially provide a temporally-optimized privileged route for olfactory reafferent input to affect the ongoing cortical activity, in line with recent reports in humans[106]. This could explain the efficacy of experimental manipulations that bias learning[107], consolidation[108], or sleep depth[109] using odor presentation during sleep which suggests that olfaction is a royal path to the sleeping brain.

**Outlook**. Finally, in light of the wide modulation of multiple circuits by breathing during quiescence, we suggest that the entrainment by breathing potentially defines functional sub-networks of neurons within and across neural circuits. To examine this hypothesis future work will be needed to carefully examine the fine temporal structure of neuronal assemblies and their modulation by the RCD and ROR copies of the breathing rhythm throughout cortical and subcortical structures, an endeavor that might uncover such functional sub-networks involved in distinct behaviors. The mechanisms responsible for the widespread pacing of limbic circuits by breathing described here might underlie the powerful role of breathing in spiritual and rehabilitation practices[110] and emotional disturbances in patients[111–113].

In summary, the data provided here suggest that breathing provides a constant stream of rhythmic input to the brain. In addition to its role as the condicio sine qua non for life, we provide evidence that the breathing rhythm acts as a global pacemaker of the brain, providing a persistent corollary discharge signal that enables the integration and segregation of information flow and processing across the distributed circuits by synchronizing local, internally-generated dynamics during offline states. In this emergent model of respiratory entrainment of limbic circuits, we speculate that this perennial rhythm not only coordinates memory consolidation dynamics during offline states but likely enables the integration of exteroceptive and interoceptive inputs and internal representations into coherent cognitive manifolds across multiple mental states.

## Methods

**Animals**. Naive male C57BL6/J mice (3 months old, Jackson Laboratory) were individually housed for at least a week before all experiments, under a 12 light-dark cycle, ambient temperature 22 °C, 50% humidity, and provided with food and water ad libitum. Experiments were performed during the light phase. All procedures were performed in accordance with standard ethical guidelines and in accordance with the European Communities Directive 2010/63/EC and the German Law for Protection of Animals (Tierschutzgesetz) and were approved by local authorities (Regierung von Oberbayern - ROB-55.2-2532.Vet_02-16-170). All efforts were made to minimize the number of animals used and the incurred discomfort.

**Surgery**. Anesthesia was induced with a combination of medetomidine (0.5 mg/kg), midazolam (5 mg/kg), and fentanyl (0.05 mg/kg), and the surgical plane of anesthesia was maintained using 1% Isoflurane in O₂. Body temperature was maintained at 37 °C with a custom heating pad. Analgesia was provided by means of subcutaneous administration of metamizole (200 mg/kg) and local subcutaneous administration of a mixture of lidocaine (5 mg/kg) and bupivacaine (5 mg/kg) during OP and meloxicam (1 mg/kg) for 7 days postoperatively. Enrofloxacin (5 mg/kg, subcutaneous administration) was also provided postoperatively. For free behavior recordings, electrode bundles, multi-wire electrode arrays, or silicon probes were implanted chronically. Recordings targeted the medial prefrontal cortex (stereotaxic coordinates: 1.7–2 mm anterior to the bregma (AP), 0.3 mm lateral to the midline (ML), and 0.8 to 1.4 mm ventral to the cortical surface (DV)), dorsal hippocampus (AP: -2.3 mm, ML: 1.5 mm, DV: 0.8–1.5 mm), V1 (AP: -3.0 mm, ML: 2.3 mm), BLA (AP: -1.7 mm, ML: 3 mm, 4 mm DV), and NAc (AP: 1.2 mm, ML: 1 mm, DV: 4 mm)[114]. For head-fixed recordings, a craniotomy above the targeted structure and a midline bilateral craniotomy above the mPFC was performed to enable the recording from all cortical layers. Dura was left intact and craniotomies were sealed with Kwik-Cast (WPI, Germany) after surgery and after each recording session. For electromyographic (EMG) and electrocardiographic (ECG) recordings, two 125 μm Teflon-coated silver electrodes (AG-5T, Science Products GmbH) were sutured into the right and left nuchal or dorsal intercostal muscles, using bio-absorbable sutures (Surgicryl Monofilament USP 6/0). Wires were connected to a multi-wire electrode array connector

(Omnetics) attached to the skull. For the recording of the neural activity of the olfactory epithelium, which was used as a proxy for respiration[29], a small hole was drilled above the anterior portion of the nasal bone (AP: + 3 mm from the nasal fissure, ML: + 0.5 mm from midline) until the olfactory epithelium was revealed. A 75 μm Teflon-coated silver electrode (AG-3T, Science Products GmBH) was inserted inside the soft epithelial tissue. Approximately 500 μm of insulation was removed from the tip of this wire and the other end was connected to the same Omnetics connector as the rest of the electrodes. Two miniature stainless steel screws (#000-120, Antrin Miniature Specialties, Inc.), pre-soldered to the copper wire were implanted bilaterally above the cerebellum and served as the ground for electrophysiological recordings and as an anchoring point for the implants. All implants were secured using self-etching, light-curing dental adhesive (OptiBond All-In-One, Kerr), light-curing dental cement (Tetric Evoflow, Ivoclar Vivadent), and autopolymerizing prosthetic resin (Paladur, Heraeus Kulzer).

**Behavior**. Electrophysiological recordings of the mice took place before and after each behavioral session in the home-cage. Home-cage consisted of clear acrylic filled with wood chip bedding and a metal grid ceiling which was removed for the purposes of the recordings. Food pellets were distributed in the home-cage and water was placed inside a plastic cup. Nesting material was available in the home-cage and utilized by the mice (typically building a nest in a corner). Exploratory behavior was recorded in a cheeseboard maze, consisting of a 60 cm diameter acrylic cylinder with a wooden laminated floor perforated with 10 mm diameter holes. For recordings of mice running freely on a wheel, a horizontal wheel (Flying Saucer) was permanently placed inside the home-cage. The mice typically exhibited long-running episodes on the wheel, with interspersed sleep episodes. Fear conditioning took place in context A, which consists of a square acrylic box (30 cm × 30 cm × 30 cm). Walls were externally decorated with black and white stripes. The box was dimly lit with white LEDs (25 lux) and white noise background sound was delivered through the walls using a surface transducer (WHD SoundWaver). The floor consisted of a custom-designed metal grid (6 mm diameter stainless steel rods) connected to a precise current source (STG4004-1.6 mA, Multi Channel Systems MCS GmbH). On day 1, mice were subjected to a habituation session in context A, during which the CS⁺ and CS⁻ (7.5 kHz, 80 dB or white-noise, 80 dB) were presented four times each. Each CS presentation consisted of 27 pips (50 ms duration, 2 ms rise and fall) with 1.1 s inter-pip interval. On the same day, in the fear conditioning session, CS⁺ was paired with the US. To serve as US, a mild electric foot-shock (1 s duration, 0.6 mA, 50 Hz AC, 5 CS-US pairings, 20–60 s randomized inter-trial intervals) was delivered to the mice through the metal grid. The onset of the foot-shock coincided with the offset of the conditioned stimulus. During the memory retrieval session, mice were presented with four CS⁻ and four CS⁺ presentations 24 h after conditioning, in a distinct context B. For experiments involving pharmacological manipulation, a second retrieval session took place 12 days after fear conditioning. For experiments involving innate fear responses, mice were exposed for 10 min to a neutral context while a small filter paper, scented with the odorant 2-methyl-2-thiazoline (2MT) (M83406-25G; Sigma Aldrich), was placed in the environment. 2MT is a synthetic odorant, chemically related to the fox anogenital gland secretion 2,4,5-trimethyl-3-thiazoline (TMT), that induces robust innate fear responses, in contrast to TMT[115]. The sequence of the experimental protocol is schematized in Supplementary Fig. 2a.

**Behavioral analysis and state segmentation**. For the purpose of behavioral state detection in freely-behaving mice, the movement of the animal was tracked using a three-axis accelerometer (ADXL335, Analog Devices) incorporated in the headstage, which was used as the ground truth for the head-motion. Accelerometer data were sampled at 30 kHz and the sensitivity of the accelerometer is 340 mV/g (g is the standard acceleration due to gravity; ~9.8 m/s²). The gravity vector is differentially reflected in the different axes as a function of posture. Since the accelerometer measures simultaneously the dynamic acceleration due to head movement and the static acceleration due to gravity, the first time derivative of the acceleration was calculated (jerk; units: g/s). This measure eliminates the effect of gravity and the dynamic acceleration dominates. The effect of gravity on the different axes is amplified during head rotations. The jerk of each axis was analyzed separately for the quantification of head-motion, however for the behavioral state detection the magnitude of the jerk was quantified as: $J = \sum_{k=1}^{3}\left|\frac{\partial \overrightarrow{a_k}}{\partial t}\right|$, where $\overrightarrow{a_k}$ is the acceleration for each axis and smoothed in time using a narrow Gaussian window (2 s, 100 ms s.d.) (Supplementary Fig. 1e). This head micromotion is not used to extract the respiration signal used throughout the manuscript, which is instead recorded from the nasal cavity. The only exception is Supplementary Figs. 1a, 2c, d, where the head-mounted accelerometer signal is also used for purely demonstration purposes, to highlight the fact that respiration is also reflected in the head micromotions.

Additionally, the activity of mice was tracked using an overhead camera (Logitech C920 HD Pro). The camera data were transferred to a computer dedicated to the behavior tracking and were acquired and processed in real-time using a custom-designed pipeline based on the Bonsai software (v2.0 - v2.6)[116]. Video data were synchronized with the electrophysiological data using network events. Video was preprocessed to extract the frame-to-frame difference and calculate a compound measure that we found provided an excellent proxy for the

behavioral state. Video frames were thresholded and binarized (Supplementary Fig. 9e–h). A logical exclusive OR operation was applied on consecutive frames, a calculation that provides the effective frame difference. The sum of these differences provides a measure of overall change between consecutive frames. We found that the changes in the amplitude of variance of this measure over time are informative for the current state of the mouse. Complete immobility is easily distinguishable using this measure, due to the low amplitude and small variance of the signal. A threshold was set manually such that even small muscle twitches during sleep were captured, but breathing-related head-motion was below the threshold. Using the density of head micromotions and muscle twitches, we were able to classify behavioral segments as active awakening, quiescence, or sleep (Supplementary Fig. 9i, j). For head-fixed recordings, we relied solely on high-resolution video of the mouse snout and body, from which we derived a micromotion signal that was used in the same way as the jerk-based signal for freely-behaving mice.

**Head-fixed recordings**. For high-density silicon probe recordings, we exploited the advantages of head-fixed mouse preparation. Large-scale neuronal recordings reported are collected with the following methods in head-fixed mice, unless otherwise noted in the figure legends. Mice were implanted with a lightweight laser-cut stainless steel headplate (Neurotar) above the cerebellum. After recovery from surgery, mice were habituated daily for 3-4 days to head-fixation prior to experimentation. A modified Mobile Home-Cage (Neurotar) device was used, enabling mice to locomote, rest, and occasionally transition to sleep, within a customized free-floating carbon fiber enclosure (180 mm diameter and 40 mm wall height). Animal behavior was monitored using two modified 30 fps, 1080p infrared cameras (ELP, Ailipu Technology Co), equipped with modified macro zoom lenses.

**In vivo electrophysiology**. LFP and single-unit activity were recorded using either 12.5 μm Teflon-coated Tungsten wire (California Fine Wire) or custom-designed silicon probes (Neuronexus and UCLA)[117]. High-density silicon probes (A1x64-Poly2-6mm-23s-160) were used for hippocampal CSD profiles, prefrontal depth profiles while multi-shank probes were used for prefrontal CSD analyses (A16x1-2mm-50-177). Individual electrodes or probe sites were electroplated to an impedance of 100–400 kΩ (at 1 kHz) using a 75% polyethylene glycol - 25% gold[118] or PEDOT solution[119]. NanoZ (White Matter) was used to pass constant electroplating current (0.1–0.5 μA) and perform impedance spectroscopy for each electrode site. A reversed-polarity pulse of 1 s duration preceded the plating procedure to clean the electrode surface. After electroplating, electrodes impedance was tested in saline (at 1 kHz) and arrays were checked for shorts. Electrodes were connected to RHD2000 chip-based amplifier boards (Intan Technologies) with 16–64 channels. Broadband (0.1 Hz–7.5 kHz) signals were acquired at 30 kHz. Signals were digitized at 16 bit and multiplexed at the amplifier boards and were transmitted to the OpenEphys recording controller using thin (1.8 mm diameter) 12-wire digital SPI (serial peripheral interface) cables (Intan Technologies). Typically 32–256 channels were recorded simultaneously using the OpenEphys GUI. Data acquisition was synchronized across devices using custom-written network synchronization code.

**Respiration measurement**. Breathing was measured using olfactory EEG recordings[29], implanted as described in the Surgery section. Following OD, the amplitude of the olfactory EEG signal was dramatically reduced. To quantify the effect of this manipulation on the neuronal entrainment by breathing, we additionally recorded the respiratory rhythm using a fast response thermistor (GLS9-MCD, TE Connectivity) placed in close proximity to the naris of head-fixed mice.

**Anatomical analysis**. After plating, electrodes and silicon probes were coated with DiI (Thermo Fischer Scientific), a red fluorescent lipophilic dye[120]. Upon insertion in the brain, the dye is slowly incorporated in the cell membranes and diffuses laterally along the membrane, allowing the visualization of the electrode track and the histological verification of the electrode position. After the conclusion of the experiments, selected electrode sites were lesioned by passing an anodal current through the electrode[121]. Typically, 10 μA current was passed for 5 s to produce lesions clearly visible under the microscope. One day was allowed before perfusion, to enable the formation of gliosis. Electrode tip locations were reconstructed with standard histological techniques. Mice were euthanized and transcardially perfused through the left ventricle with 4% EM grade paraformaldehyde (PFA) (Electron Microscopy Sciences) in 0.1 M PBS. Brains were sectioned using a vibratome 50–80 μm thick sections) and slices were stained with DAPI and mounted on gelatin-coated glass microscopy slides.

**LFP analysis**. Raw data were converted to binary format, low pass-filtered (0.5–400 Hz) to extract the local field potential component (LFP), and down-sampled to 1 kHz. LFP signals were filtered for different frequency bands of interest using zero-phase-distortion sixth-order Butterworth filters. All data analysis was performed using custom-written software. Neuroscope data browser was used to aid with data visualization[122].

**Spectral analysis**. LFP power spectrum and LFP–LFP coherence estimations were performed using multitaper direct spectral estimates. For respiration frequency analyses, data were padded and a moving window of 3 s width and 2.4 s overlap was applied to the data. Signals were multiplied with two orthogonal taper functions (discrete prolate spheroidal sequences), Fourier transformed and averaged to obtain the spectral estimate[123]. Magnitude-squared coherence was calculated using these multitaper direct spectral estimates. For gamma frequency analyses, a window of 100 ms with 80 ms overlap, and four tapers were used. For some analyses and examples, to obtain a higher resolution in both time and frequency domain, data were transformed using complex Morlet wavelets (bandwidth parameter: 3, center frequency: 1.5). Convolution of the real and imaginary components of the transformed signal enables the extraction of the instantaneous amplitude and phase of the signal for each scale. For some example signal visualizations, we found it useful to utilize the real part of the wavelet transformed signal, which preserves both phase and amplitude information (Fig. 8a). For the power comodulation analysis[124], the instantaneous multitaper estimate of the spectral power time series for each frequency bin in each structure was calculated and the Spearman correlation coefficient of every pair was calculated. To characterize the causal relationship between the respiratory signal and the prefrontal LFP, spectrally resolved Granger causality was calculated using the multivariate Granger causality toolbox[125]. Briefly, unfiltered LFP traces were detrended and normalized. The order of the vector autoregressive (VAR) model to be fitted was calculated using the Akaike information criterion. To correct for the effect of SNR differences on Granger causality estimates, Granger causality was calculated for both the original and the time-reversed signals and compared[126] (Supplementary Fig. 1i).

To establish the phase shift between two different signals, we used two different approaches, one in the time and the other in the phase domain. In the time domain (e.g., Supplementary Fig. 1b left), we detected the troughs or peaks of one signal (using a local minima detection algorithm on the band-pass filtered signal) and used these times to calculate the triggered average of the second signal. For the phase domain analyses (e.g., Supplementary Fig. 1b right), we calculated the instantaneous phase of each of the signals (as the four-quadrant inverse tangent of the analytical signal calculated from the Hilbert transform of the narrow-band-pass-filtered signal) and plotted the distribution of point-to-point phase differences.

**Phase modulation analysis**. For phase analyses, the signal was filtered in the desired narrow frequency band and the complex-valued analytic signal was calculated using the Hilbert transform $\rho(t) = e^{-i\phi(t)}$. The instantaneous amplitude at each time point was estimated based on the vector length, while the instantaneous phase of the signal was computed as the four-quadrant inverse tangent of the vector angle. A phase of 0° corresponds to the peak of the oscillation and a phase of 180° to the trough of the oscillation. The waveshape of the respiratory signal and its LFP counterparts are highly asymmetric, resulting in nonuniform phase distribution of this reference signal (Supplementary Fig. 3c). This deviation from uniformity is catastrophic for the phase modulation statistics since it biases the phase-detection leading to false-positive results. To account for this potential bias, the circular ranks of the phase distribution were computed and the phase distribution was transformed using the inverse of the empirical cumulative density function (ECDF) to return a signal with a uniform prior distribution. After this correction, the phases can be assumed to be drawn from a uniform distribution enabling the unbiased application of circular statistics[30,57,127]. Point-processes with <200 events in the periods of interest were excluded from phase analyses, due to the sample-size bias of these analyses[57]. For the quantification of phase modulation, the variance-stabilized log-transformed Rayleigh's test $Z$ ($log(\frac{R^2}{n})$), where $R$ is the resultant length and $n$ the sample size, log is natural logarithm) was used[30,57,127]. This statistic quantifies the nonuniformity of a circular distribution against the von Mises distribution. Since ECDF transformation nonlinearly distorts the phase, non-corrected phase samples were used for characterizing the preferred phase.

**Phase-amplitude cross-frequency coupling**. For power-phase cross-frequency coupling, the modulation index (MI), as well as the mean resultant length (MRL), was calculated for each phase and amplitude pair[128]. The phase was evaluated for 1–20 Hz with a bandwidth of 1 Hz and a step of 0.2 Hz using the Hilbert transform and correction for nonuniformity as described above. The amplitude was evaluated for 20–120 Hz with 5 Hz bandwidth and 3 Hz step. Shuffling statistics were used to evaluate the statistical significance of the MI and MRL by shuffling the phase and amplitude values.

**Current-source density analysis**. Current-source density analysis was performed using the inverse CSD method[129] with activity diameter 1 mm for slow and 0.5 mm for fast network events, 0.05 s.d., smoothed using varying cubic splines and extracellular conductivity σ = 0.3 S/m based on calculations of isotropic and ohmic tissue impedance[130,131]. Importantly, all results were qualitatively confirmed by exploring the parameter space as well as using the classic second derivative CSD estimation method[132]. Occasional malfunctioning recording sites were interpolated from neighboring sites and all relevant sinks and sources were characterized and quantified from portions of data with no interpolated sites.

**Layer assignment**. For the hippocampal high-density silicon probe recordings, channel layer assignment was performed based on established electrophysiological patterns of activity for different laminae[8]. The middle of the pyramidal layer was assigned to the channel with the highest amplitude of ripple oscillations (100–250 Hz band) and associated spiking activity. Neurons recorded dorsal of the channel with the highest SWR power were characterized as deep CA1 pyramidal neurons[51]. Conversely, neurons recorded ventrally of this reference channel were characterized as superficial CA1 pyramidal neurons. Given that neuronal spikes can be identified in more than one channel of the polytrode, neurons were assigned to the channel with the highest spike amplitude[133]. Well-described CSD profiles of hippocampal oscillatory patterns were used to assign somatodendritic CA1 and DG layers to channels (Supplementary Fig. 5e). The middle of stratum radiatum was assigned to the channel with the deepest sharp-wave current-source density sink associated with ripple oscillations[134,135]. Stratum oriens was defined as the channels above the pyramidal layer SWR CSD source and below the internal capsule, characterized by a positive component of the sharp waves. For the identification of DG layers, we used the CSD and amplitude versus depth profile of dentate spikes (DS)[52]. DS are large-amplitude events that occur naturally during offline states and reflect synchronized bursts of medial and lateral entorhinal cortex[52]. The outer molecular layer was defined as the Type-I dentate spike (DSI) sink, while the middle molecular layer was assigned as the channels exhibiting DSII sinks. The inner molecular layer was defined as the channel of the deepest secondary sink in the SWR triggered CSD, which is ventral of the DSII middle molecular layer sink. The source of DSII spikes, which corresponds to a typically more localized source preceding SWR events[135], together with the polarity reversal of the DSII, which occurs above the granule cell layer[52], enables the precise detection of this layer[136]. Stratum CA1 lacunosum-moleculare was defined as the difference between the theta-trough triggered CSD sink and the outer molecular DSII sink. This corresponds to approximately the dorsal third of the theta sink. For V1 CSD, layers were assigned based on known profiles during visual stimulation. An early sink following visual stimulation between layers 3 and 4[137], a later sink between layers 5 and 6, and a peak of the high-frequency power in mid-layer 5[138,139].

**Network event detection**. Ripples were detected from a CA1 pyramidal layer channel using the instantaneous amplitude of the analytic signal calculated from the band-pass filtered (80–250 Hz). The instantaneous amplitude was referenced to the amplitude of a channel typically from the cortex overlying the hippocampus, was convolved with a Gaussian kernel (100 ms, 12 ms s.d.) and normalized. The mean and s.d. of the referenced amplitude were calculated for periods of quiescent immobility and slow-wave sleep. Ripples were detected as events exceeding 3 s.d with a minimum duration of four cycles and were aligned on the deepest trough of the band-pass filtered signal. Gamma bursts were detected using a similar procedure, but for the relevant frequency band and behavioral states. Dentate spikes (DS) were detected as large deviations (>3 s.d.) of the envelope of the 2–50 Hz band-pass filtered LFP signal from the DG hilar region (using the channel from the hilar region with the peak amplitude), referenced to the CA1 pyramidal layer. DS were detected from the subtracted envelopes of the two channels. The trigger time for the calculation of the DS CSDs was the peak of this band-pass amplitude. Following detection, DS were clustered in two types using k-means clustering on the 2D space defined by the two principal components of the CA1/DG depth profile of each spike. The classification between the two types was based on the PCA of the translaminar LFP profile of the detected events. UP and DOWN states were detected by binning the spike train for every single-unit in 10 ms windows, normalized, and convolved with a 0.5 s wide, 20 ms s.d. Gaussian kernel. The average binned spike histogram was calculated across all simultaneously recorded cells (including PNs and INs). DOWN states were detected as periods longer than 50 ms with no spikes across all the cells and the exact onset and offset of DOWN states were detected. UP states were detected as periods contained between two DOWN states, lasting between 100 and 2000 ms, with the average MUA activity during this period exceeding the 70th percentile of the MUA activity throughout the recording.

**Single-unit analysis and classification**. Raw data were processed to detect spikes and extract single-unit activity. Briefly, the wide-band signals were band-pass filtered (0.6–6 kHz), spatially whitened across channels and thresholded and putative spikes were isolated. Clustering was performed using template matching algorithms implemented in Kilosort2[140] and the ISO-SPLIT method implemented in MountainSort package[141] and computed cluster metrics were used to pre-select units for later manual curation. Specifically, only clusters with low overlap with noise (<0.05), low peak noise (<30), and high isolation index (>0.9) were considered for manual curation, using custom-written software. At the manual curation step, only units with clean interspike interval (ISI) period, clean waveform, and sufficient amplitude were selected for further analysis. For the data collected with high-density polytrodes, after manual curation, a template of the spike waveform across 10 geometrically adjacent channels was calculated and the unit was re-assigned to the channel with the largest waveform amplitude. To classify single-units into putative excitatory and inhibitory cells, a set of parameters based on the waveform shape, firing rate, and autocorrelogram were calculated. The two parameters that

offered the best separation, in accordance with what has been reported in the past, were the trough-to-peak duration[35] and the spike-asymmetry index (the difference between the pre- and post- depolarization positive peaks of the filtered trace)[142], reflecting the duration of action potential repolarization which is shorter for interneurons[143,144] (Supplementary Fig. 3a, b). Single-units with <200 spikes in the periods of interest were excluded from all analyses. Spike trains were z-scored across all periods analyzed (typically quiescence and sleep periods unless otherwise specified). Dimensionality-reduction was performed on the inspiration-triggered activity and correlation matrix using PCA. Isomap and other dimensionality techniques were implemented using the Matlab Toolbox for Dimensionality Reduction. Importantly, the results were qualitatively independent of the particular dimensionality-reduction method used (Supplementary Fig. 4c). Co-firing index for each pair of units was defined as the mean ratio of co-occurring spikes to the total number of spikes for both units for each 10 ms bin (Fig. 8f). For the ripple and opto-ripple-evoked response analyses, triggered spike-trains were normalized across events, neurons, and breathing phase, in order to account for intrinsic firing rate differences across cells and breathing phase (Figs. 8e, i, 9i and Supplementary Fig. 8l).

**Optogenetics**. For the generation of opto-ripples, mice were unilaterally injected with 300 nL of AAV2/9-CaMKIIa-ChETA(E123T/H134R)-eYFP-WPRE.hGH in the dCA1 region (AP: −2.3 mm, ML: 1.5 mm, DV: 1.2) using positive pressure through glass pipettes (tip diameter 20–30 μm) connected to a Picospritzer. Three weeks after the injection, the functional effect of the opsin was tested in the awake head-fixed mouse using electrophysiological recordings from the dCA1 using a silicone probe coupled with a 200 μm diameter optic fiber. 465 nm light was delivered using LEDs (Plexon) and the intensity was calibrated to generate opto-ripples (Fig. 9c) of amplitude similar to the intrinsically recorded oscillation (typically 25–50 mW/mm²). The stimulation pattern was selected to be a half-sine wave with 200 ms duration in order to generate opto-ripples[56,73] with duration and characteristics similar to the intrinsic ripples. After this process, an optic fiber was chronically implanted at the same coordinates and electrophysiological recordings from the prefrontal cortex were performed while generating opto-ripples in the dCA1. Stimulations were delivered with random inter-stimulation intervals (1–3 s) and were later analyzed as a function of the respiration phase during which they were delivered.

In these experiments, the optogenetic induction of the ripples was performed by stimulating the dCA1 region using a single smooth half-sine wave (200 ms duration; 1–3 s ITI). As a result of this stimulation, the local hippocampal circuit oscillates in a ripple-like fashion following the ING/PING mechanism underlying endogenous ripples[73]. The frequency of the generated ripples is thus not imposed externally, but it is rather generated natively in the circuit. This somewhat lower frequency of the optogenetically induced ripples agrees with other reports in the literature[56,73], though the exact reasons for this are not yet understood. Based on theoretical work[145] the exact frequency of the oscillation depends on the balance of the drive of excitatory and inhibitory populations, which clearly is different between endogenous sharp-wave-associated Schaffer-collaterals input, that targets both pyramidal cells and interneurons, and optogenetic excitation that targets pyramidal cells due to CaMKIIa-driven opsin expression. However, these artificial ripples appear to be qualitatively and functionally related to the intrinsic ripples, given their capacity to enhance behavior and learning and to extend the neuronal ensemble content of intrinsic events[56,146].

**Pharmacology**. To causally prove the role of respiratory epithelium neurons in driving oscillations in the prefrontal cortex of mice, we induced selective degeneration of the olfactory epithelium cells (Supplementary Fig. 7a) by systemic administration of methimazole[65]. Mice were injected intraperitoneally with methimazole (75 mg/kg). No generalized effect on weight, activity patterns, or gross anatomy of the mPFC, OB, or hippocampus were detected following the treatment (Supplementary Fig. 7l–p). The effect on neuronal dynamics was characterized at 3, 7, or 10 days following the ablation of OSNs, with no appreciable differences between these time points.

**Statistical analysis**. For statistical analyses, the normality assumption of the underlying distributions was assessed using the Kolmogorov–Smirnov test, Lilliefors test, and Shapiro–Wilk tests. Further, homoscedasticity was tested using the Levene or Brown–Forsythe tests. If the tests rejected their respective null hypothesis non-parametric statistics were used, alternatively, parametric tests were performed. When multiple statistical tests were performed, Bonferroni corrections were applied. Where necessary, resampling methods such as bootstrap and permutation tests were used to properly quantify significance. For box plots, the middle, bottom, and top lines correspond to the median, bottom, and top quartile, and whiskers to lower and upper extremes minus bottom quartile and top quartile, respectively.

**Reporting summary**. Further information on research design is available in the Nature Research Reporting Summary linked to this article.

## Data availability

Relevant data that support the findings of this study are available from the authors upon reasonable request. Source data are provided with this paper.

## Code availability

Acquisition, processing, and analysis was performed using publicly available software as described in the "Methods" section. Custom code used to analyze this data is available online (https://github.com/nikolaskaralis/Karalis2021_NatureComm).

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

## Acknowledgements
We thank E. Blanco Hernandez for input and help with histological verification of olfactory deafferentation, G. Schwesig, and E. Resnik for valuable input, R. Ahmed for technical assistance, J. Lu for assistance in the experiments, S. Masmanidis for providing silicon probes, and all the members of the Sirota laboratory for helpful discussions and comments on the manuscript. This work was supported by grants from Munich Cluster for Systems Neurology (SyNergy, EXC 1010), Deutsche Forschungsgemeinschaft Priority Program 1665 and 1392 and Bundesministerium für Bildung und Forschung via grant no. 01GQ0440 (Bernstein Centre for Computational Neuroscience Munich), European Union Horizon 2020 FET-POACT program via grant agreement no.723032 (BrainCom) (AS), EMBO Long-Term Fellowship ALTF 914-2018, and European Union's Horizon 2020 research and innovation program under the Marie Skłodowska-Curie grant agreement No. 843236 (NK).

## Author contributions
N.K. and A.S. designed the experiments and data analysis, interpreted the data, and wrote the manuscript. N.K. performed the experiments and analyzed the data.

## Funding

## Competing interests
The authors declare no competing interests.
