## [Peer Review File · Nature Communications]

Breathing coordinates cortico-hippocampal dynamics in mice during offline statesREVIEWER COMMENTS

Reviewer #1 (Remarks to the Author):

Synchronized and coordinated activity is essential for neural network function. Breathing has been recognized to provide a global oscillatory signal to entrain the neural activity across widespread brain regions. However, the source of the respiration-related rhythm in the brain is not fully understood. In addition to nasal breathing-entrained olfactory signals, Karalis and Sirota identified the efference copy of the brainstem respiratory rhythm as another source. Using large-scale recordings from multiple cortical and subcortical brain regions, the authors first showed that breathing entrains the prefrontal activity (low-frequency LFPs, gamma oscillations, and single-unit firing) across different behavioral states (especially the offline states), and then extended the analysis to other brain regions: hippocampus, thalamus, BLA, NAc and V1. Pharmacological ablation of the olfactory epithelium eliminated respiration modulation of LFPs but not of neuronal firing. Inter-regional analysis and optogenetic perturbations indicated that that breathing rhythm couples hippocampal sharp-wave ripples and modulates cortical DOWN/UP state transitions. Overall, the manuscript contains a set of elegant experiments and analysis with high-quality figures and clearly-written text. Although a few very recent papers (e.g. Mofleh and Kocsis (2021) and Girin et al. (2021) published in Scientific Reports) also reported that respiration entrains the neural activity in the olfactory, prefrontal and hippocampal circuits in quiet awake and sleep states, the current study provides the most comprehensive and in-depth analysis on the source of respiration rhythm in the brain. I have only minor suggestions.

In Fig. 2k, add abbreviations for the brain structures.

In Fig. 3a, it would be helpful to add a horizontal line as in Fig. 2b to indicate the significance threshold for the logZ.

For Fig. 5g, t-test does not seem appropriate since the data points are not normally distributed. Suggest using the non-parametric alternative Wilcoxon signed-rank test instead. For Fig. 5l, if the data points are not normally distributed, the non-parametric test should be used.

In Fig. 6i, suggest not using red to mark the DOWN state in the mPFC LFP trace, as red is used for UP state in other panels in the same figure.

P11 line 259-260. "...mice exhibited intact memory and fear expression, suggesting that the RCD might be underlying the behavioral expression". An alternative interpretation is that fear memory and expression do not depend on respiration, rather the 4 Hz may be a byproduct of the freezing behavior.

Page 12 line 291-292, "UP and DOWN state modulation was not affected by olfactory deafferentation, suggesting that RCD is the source of this modulation". This may be an overstatement. It is possible that olfactory inputs contribute to this modulation and RCD provide a redundant source. When the OE is ablated, RCD can compensate the loss of olfactory inputs.

In Fig. 8, the UP and Down mark the state onset, but it is not clear where the start is. They do not seem to coincide with higher and lower firing rates in the cortex.

Reviewer #2 (Remarks to the Author):

Summary

The authors found that localized neural delta oscillations and gamma bursts in pre-frontal cortex, nucleus accumbens and basolateral nucleus of the amygdala as well as hippocampal ripples are locked to breathing rhythm. Furthermore, they showed that spiking activity of prefrontal cortex, visual cortex, midline and sensory thalamus, nucleus accumbens and basolateral nucleus are phase-locked to breathing cycles. Next, they showed that after olfactory deafferentation the locking of neuro-oscillations in prefrontal cortex to the breathing cycles is interrupted and spiking activity is more disconnected from breathing. Lastly, using optogenetics, they show that spiking activity in the nucleus accumbens and prefrontal cortex is modulated by both ripples and breathing cycle.

Presentation and Structure

The overall structure of the data presentation is well done, and the overall concept, experimental

strategy and discussion are somewhat easy for the reader to follow. A potential point of improvement would be to improve the description of experimental set ups for each procedure (or figures), improve the arrangement of figures and graphs, even though I do understand that the study contains both a descriptive and an experimental part with some inconsistencies and unavoidable overlaps. Lastly, the outlook addresses some general concepts in a manner that is quite superficial and overambitious where it is difficult to understand what the authors' take on their finding and their message to the field is, as for instance, a more detailed look into specific behaviours controlled by medial prefrontal cortex (see below).

Major points

- Explanations of the experimental setups are at times not very clear:

- How was "quiescence" defined? Did the authors differentiate between quiet wakefulness and sleep (with NREM and REM)?

- Do Fig. 2-4 all belong to data from the same mice? Can you briefly introduce the new experiment after the one for Fig.1?

- Why were some mice head-fixed others not for Fig. 2-4?

Could the authors explain explicate the experimental setups more clearly in the text and "walk the reader through the process" in more details? Could the authors also provide simple schematics in each figures where a new experimental procedure (or animal preparation, e.g., tetrode vs (multi-)shank recordings) is introduced that show the whole setup in terms of where electrodes are? It appears important to also clarify whether if the mice are freely moving head-fixed for each result sections. Accordingly, the authors should explain why some mice were head-fixed, others not.

Could the authors also explain how behaviour was scored, how the different stages were differentiated from each other and give some exemplary graphs and figures of the process?

- Lines 200-201, 251 of the main text: the authors mention gamma rhythms in the PFC are generated through input from the olfactory bulb, but they only show the phase-amplitude coupling of gamma to respiratory cycles (both in Fig.4 and Fig.5). However, it might also be possible that gamma bursts are regulated through the delta frequency in PFC and this establishes the link between gamma bursts and breathing cycles. Consequently, the lower modulation index could be due to the decoupling of delta and breathing rhythm. Could the authors provide additional gamma burst analyses (i.e. power, duration and occurrence rate of gamma bursts before and after deafferentation) to support their claim?

- The chosen frequency of the optogenetically induced ripples is comparatively low (70-250 Hz) for mice (a frequency range from 100Hz-250Hz is usually found in the literature. Ripples in rats are often described to have a lower range). Indeed, the authors themselves show that the spontaneous ripples they detected have a frequency range around 150Hz. Could the authors explain why they chose a lower frequency range? And could they provide some descriptive analyses of their induced ripples, spontaneous ripples and perhaps high gamma bursts (i.e. average power, peak frequency, duration, number of cycles, symmetry of the event)? That would strengthen the authors' optogenetic experiments and support their claims.

- About general aspects of the experiment inducing ripples with optogenetics: Could the authors explain the rationale of the experiment and clarify the interpretation of the results? Right now, it is not clear in the manuscript what the authors try to show with this experiment: they induce ripples and lock them to breathing cycle and they see similar activity in PFC as in spontaneous ripples. However, how does this replication of an observed pattern show that it really is breathing that causally modulates coupling of hippocampus and prefrontal cortex? What would we see in the experiment if breathing was not the synchronizing cause? If possible, could the authors redo the analysis in Fig.7s with the data of the deafferentation experiment? This would actually support what they want to claim with the optogenetics experiment.

- About the outlook in general: The statement about the default mode network seems a bit out of place. Even though the authors showed that breathing influences several remote brain regions, I would still be hesitant to link it to a quite specific overall brain state, especially since it is rather mentioned in the context of MRI measurements and less so in the electrophysiological field. At the same time, since breathing is not only linked to LFP and spiking as described but to several physiological features of the entire organism (i.e. blood pH, emotional state, cardiovascular state via brain stem influence) such a statement could also be dismissed as rather generic as chances

are high that breathing could have an indirect influence on the default mode network via effects on other organs.

In the beginning of the study, the authors looked at breathing during fear. It would be much more compelling if the authors gave thoughts on possible implications of breathing influence on medial PFC function during fear behaviour as this brain region has been described extensively in this context. They could transfer the paragraph from lines 402-411 to the outlook.

Minor points

- Fig. 2g: Could the authors explain how the power was normalized?
- Fig. 2i-j: What does time point zero exactly refer to?
- Fig. 2h: Could the authors mention that it is the spike-phase modulation that is shown here?
- Fig. 4b: Could the authors address the fact PFC shows a greater phase-amplitude coupling than OB? How could that be? Please comment and discuss this finding.
- Title of Fig.5: Title says that there is no neural entrainment, however 5h-5j seem to show that it is the case and the main text also states that it does. Please clarify.
- Fig. 7s: Please clarify how PFC activity was normalized.
- Fig.7: Could the authors provide the full name of the viral construct used in the main text and a simple schematic in the figure to support the reader?
- Supp. Fig.7: Could the authors also show anatomical pictures of olfactory bulb and prefrontal cortex to show that these regions were not affected by the injections?
- Supp. Fig. 1:
 - Please briefly describe what a Thermistor is.
 - Panel i: Could the other briefly mention what SNR stands for and give a short interpretation of the panel?
- EOG is commonly understood in the field as "electro-oculogram". Please change the abbreviation (e.g. Olfactory EEG)?
- lines 125-134 of main text: the authors explain the phase-modulation with poly-synaptic projections. Since the modulation follows a dorso-ventral gradient, is it possible that there is after all a possible mechanical influence due to the air-flow through the nasal cavity which is situated right below this part of the brain? Please address this point in the discussion.
- Lines 145-176 of main text: in the last part of the paragraph the entorhinal input to dentate gyrus is mentioned. Could the authors briefly explain why those inputs are important here? Please cite some helpful sources for non-expert readers who don't know the specific anatomy of the hippocampal formation. Please discuss these possible connections.
- Lines 256-258 of main text: Could the authors cite some sources at this point and make clear that this circuit has already been proposed? It comes across as if the authors came up with the term.
- Lines 277 of main text: the authors mention an observed current sink in medial entorhinal cortex in sup. Fig. 8c, d but the description of the figure says that it was recorded in dorsal Hippocampus.
- Lines 299-300 of main text: Can the authors cite sources for the claim of "RCD-mediated inputs to DG"?
- Lines 379-385 of main text: Could the authors mention some sources for interested readers?

Reviewer #3 (Remarks to the Author):

In this article, the authors record from several brain areas of the mouse brain, both local field potential and single-unit activity, to gather evidence related to the hypothesis that breathing serves as a brain-scale coordinator of neural activity. They center their efforts in the so-called limbic system but record from other areas as well. Also, the work focuses on what the authors call "off-line" behavioral states, i.e., periods of inactivity and sleep.

The authors found widespread respiratory modulation of LFPs and single-unit activity across brain areas, consistent with previous findings. They also characterize the incoming synaptic volleys associated with respiratory modulation through current source density analysis. In addition, they studied the role of this modulation in hippocampal-cortical communication and the respiratory modulation of cortical UP/DOWN states. Finally, to address the origin of the modulation, they intervened animals by eliminating the input to the brain from the olfactory epithelium (i.e.,

deafferentation), which is mechanically stimulated in each respiratory cycle and relays this activity to the olfactory bulb. The deafferentation effectively abolished most LFP and a fraction of the single-unit modulation. However, some single-unit respiratory modulation survives OB deafferentation, strongly suggesting a corollary discharge from the respiratory brainstem.

The work is well-executed, the evidence is compelling, and the terrain covered is vast. Several simultaneous and different sources of evidence are provided, including behavior, electrophysiological recordings, anatomical/histological, and interventions (pharmacology, optogenetics). Evidence of this quality is relevant for the field and can contribute to understand some of the big-picture puzzles in neuroscience.

I do have, however, some observations, comments, and requests for the authors.

1) There is a lack of clarity regarding the actual behavioral states or periods used to get the electrophysiological data. For example, it was not always clear when the data came from quiescent periods, when from sleep, or if quiescent and sleep periods were pooled together. Are both states equally represented in the data? Except for one or two plots, this is not clear. A diagram or plot containing the relative amounts of data from each period should be provided separately for each animal. I'd also include here labeling of the head-fixed data.

1.1) Also, what is the justification for pooling (at least conceptually) data from quiescent periods and sleep?

1.2) I'm also curious, why call these states "offline" without quotation marks? Is this a technical term in use in the sleep research field? I'd explain a bit about this.

2) I would like to see a detailed overview of the numbers of neurons modulated and non-modulated, per area and animal. I'm thinking of a table, or figure/diagram, containing the brain areas as columns and individual animal subjects as rows, with the number and percentage of modulated neurons as entries. Getting a good sense of the variability/heterogeneity of the responses, both across areas and especially across animals, is as informative of the phenomenon as the "population average" message. I see the inter-subject variability as crucial data, not as noise. In our times of replicability problems and of open science approaches, data like these are a must. If space is an issue, I think this is much more informative for researchers than example traces and/or anatomical diagrams.

3) I think all the data presented in boxplot format, given that it comes from small samples (e.g., $n=6$, $n=7$), should instead be provided as individual jittered data points, plus a bar indicating the mean or median. For both supplementary and paper figures. In methodologically complex studies as this, I understand small samples. Precisely because of this, I think data should be more visible to get a better sense of the evidence gathered.

4) Can you provide more information about the methimazole treatment?

- the general state of the animals after treatment (ie, sleep vs awake profiles, weight time series, feeding amounts, etc)?
- a more long-term respiratory data after treatment?

Given that it is not a widely used intervention, at least in neuroscience, and that it is a systemic perturbation, it would be useful to get a sense of the general impact of this intervention.

5) I think statements like "we identified an intracerebral centrifugal respiratory corollary discharge" or "identified a novel global mechanism" may be a bit of an overstatement. The finding is clear, and there are not too many options for the origin of the signal after deafferentation, but the data has not properly identified the pathway.

COMMENTS ABOUT FIGURES AND OTHER POINTS:

FIG 1

-1a It would benefit from showing an inhalation-triggered area LFP mean, compared with a null

model constructed by taking the trigger from random positions in the time series.

- 1b. Top signal ("Motion"). The caption doesn't refer to it.
- 1b. The spectrogram shows a peak of activity right at 4Hz. But in Fig 1c 1d & 1e peak is at 3 Hz. Why is this?
- I'd include some variability metric for all the spectra (example g, i).

FIG 2

- 2a: Mean responses need a variability metric.
- 2c: legend says "horizontal line"; should be "vertical"
- Many anatomical structures lack full naming (e.g., MO, VO, IL, DP, etc.)
- 2k: a cross indicating axes (ap, dv) would be of help here.

FIG 3

- 3d: I'd change the color of the star and circle, as they are hard to see.

FIG 5

- 5l: for completeness, I'd add the MEC LII sink comparison as well.

FIG 6

- 6f: Letter F is uppercase
- 6m,n: why including the histogram example only for n and not for m?
- 6o,q: why/how is probability expressed as stdev in these plots?

FIG7

- 7a,m What information exactly do we get by looking at the real part of the transform?

OTHER

- lines112-113: There are newer references to the relationship between ob lfp and respiration during active behavior
- line139: "both Ca1 PNs and INs..." confusing
- discussion: I think the work on "breathing as binder of orofacial sensation" (Kleinfeld et al 2014) may be relevant to discuss here.

Analyses and figures added

1. Fig. 5g: Analyzed the mPFC - OB gamma burst correlation before and after OD
2. Fig. 7s: Calculated ripple evoked mPFC depolarization after OD
3. Supplementary Fig. 1h: Calculated respiration frequency across states
4. Supplementary Fig. 1i: Added inhalation-triggered LFP, compared with a null model
5. Supplementary Fig. 3m: Calculated the mPFC DV LFP profile after OD
6. Supplementary Fig. 6: Calculated the mPFC gamma power before and after OD
7. Supplementary Fig. 7l Provided the relative weight change of mice after OD
8. Supplementary Fig. 7m: Analyzed the changes in activity and sleep of mice after OD
9. Supplementary Fig. 7n-o: Established the lack of gross anatomical differences in the OB, mPFC, and hippocampus of mice after OD
10. Supplementary Fig. 8f: Estimated dCA1 neuron evoked response to optogenetic stimulation
11. Supplementary Fig. 8g: Calculated the joint distribution of frequency and power of optogenetically evoked ripples
12. Supplementary Fig. 8h: Calculated the distribution of number of cycles of optogenetically evoked ripples
13. Supplementary Fig. 9a-d: Added schematics of recording configurations for freely behaving and head-fixed mice
14. Supplementary Fig. 9e-h: Included the process of behavior classification
15. Supplementary Fig. 9i-j: Included the process of behavior classification
16. Supplementary Fig. 9k: Calculated the relative state ratio for each subject
17. Supplementary Fig. 9l: Calculated the fraction of modulated cells by area for each subject

Figures changed

1. Fig. 1g: Boxplot replaced
2. Fig. 1i: Boxplot replaced
3. Fig. 3a: Horizontal line of significance added
4. Fig. 5g: Boxplot replaced
5. Fig. 5l: Boxplot replaced
6. Fig. 5l: Changed ratio of MEC to LEC sink
7. Fig. 6h: Boxplot replaced
8. Fig. 6i: Replaced color code of delta wave
9. Fig. 6m: Included histogram
10. Fig. 7k: Added information on the injected virus
11. Fig. 8: Marked UP/DOWN state onset
12. Supplementary Fig. 7h: Boxplot replaced
13. Supplementary Fig. 7i: Boxplot replaced
14. Supplementary Fig. 7k: Boxplot replaced

Summary of text changes

In addition to a number of changes throughout the text, we have performed the following changes as suggested by the reviewers.

1. Clarified throughout the results the experimental conditions used.
2. Clarified the behavioral classification approach used.
3. Provided background information about the hippocampal inputs and the potential role of their modulation by breathing.
4. We have expanded the discussion and references about the brainstem circuits generating the respiratory rhythm.
5. We have included newer citations, in particular in reference to the OB LFP.
6. We have included a new section discussing in more detail the potential role of respiratory entrainment in the context of fear behaviour.
7. Toned-down statements regarding the mechanism, as suggested by Reviewer 3.
8. Provide source data for all figures

Reviewer #1

Synchronized and coordinated activity is essential for neural network function. Breathing has been recognized to provide a global oscillatory signal to entrain the neural activity across widespread brain regions. However, the source of the respiration-related rhythm in the brain is not fully understood. In addition to nasal breathing-entrained olfactory signals, Karalis and Sirota identified the efference copy of the brainstem respiratory rhythm as another source. Using large-scale recordings from multiple cortical and subcortical brain regions, the authors first showed that breathing entrains the prefrontal activity (low-frequency LFPs, gamma oscillations, and single-unit firing) across different behavioral states (especially the offline states), and then extended the analysis to other brain regions: hippocampus, thalamus, BLA, NAc and V1. Pharmacological ablation of the olfactory epithelium eliminated respiration modulation of LFPs but not of neuronal firing. Inter-regional analysis and optogenetic perturbations indicated that that breathing rhythm couples hippocampal sharp-wave ripples and modulates cortical DOWN/UP state transitions. Overall, the manuscript contains a set of elegant experiments and analysis with high-quality figures and clearly-written text. Although a few very recent papers (e.g. Mofleh and Kocsis (2021) and Girin et al. (2021) published in Scientific Reports) also reported that respiration entrains the neural activity in the olfactory, prefrontal and hippocampal circuits in quiet awake and sleep states, the current study provides the most comprehensive and in-depth analysis on the source of respiration rhythm in the brain. I have only minor suggestions.

We thank the reviewer for the accurate summary and positive evaluation of our work.

In Fig. 2k, add abbreviations for the brain structures.

We have added the abbreviations in the figure legend.

In Fig. 3a, it would be helpful to add a horizontal line as in Fig. 2b to indicate the significance threshold for the logZ.

We have now added the horizontal line.

For Fig. 5g, t-test does not seem appropriate since the data points are not normally distributed. Suggest using the non-parametric alternative Wilcoxon signed-rank test instead. For Fig. 5l, if the data points are not normally distributed, the non-parametric test should be used.

We thank the reviewer for pointing this out. We have now performed non-parametric tests where applicable, including for the respective figures (5g, 5l).

In Fig. 6i, suggest not using red to mark the DOWN state in the mPFC LFP trace, as red is used for UP state in other panels in the same figure.

We thank the reviewer for noticing the inconsistency in the color code. We replaced the red with blue color in this trace.

P11 line 259-260. “...mice exhibited intact memory and fear expression, suggesting that the RCD might be underlying the behavioral expression”. An alternative interpretation is that fear memory and expression do not depend on respiration, rather the 4 Hz may be a byproduct of the freezing behavior.

Indeed, this is a correct observation. We have now replaced this sentence with the following: “Interestingly, following OD, mice exhibited intact memory and fear expression, suggesting that freezing behavior does not rely on the ROR input.”

Further, we have included the following discussion of this aspect:

“Importantly, although prefrontal 4 Hz LFP oscillations originate in fear-associated enhanced breathing, the ROR is not necessary for the expression of innate or conditioned fear behavior, in agreement with a recent report (Moberly et al., 2018). This suggests the independence of fear expression from the respiratory entrainment or the potential sufficiency of RCD for the expression of fear behavior. Consistent with such mechanism, the optogenetic induction of 4 Hz oscillations in prefrontal circuits is sufficient to drive fear behavior in naïve animals (Karalis et al., 2016), raising the possibility that this effect is mediated by the bidirectional interaction of

prefrontal networks with the respiratory centers via top-down projections to periaqueductal gray and the ascending feedback via RCD giving rise to system-level resonance at breathing frequency.”

Page 12 line 291-292, “UP and DOWN state modulation was not affected by olfactory deafferentation, suggesting that RCD is the source of this modulation”. This may be an overstatement. It is possible that olfactory inputs contribute to this modulation and RCD provide a redundant source. When the OE is ablated, RCD can compensate the loss of olfactory inputs.

We thank the reviewer for bringing this aspect to the discussion. Indeed, this argument is exactly in line with our description and interpretation of the mechanism. We recognize that this statement could be improved, we have thus replaced it with the following statement:

“In line with the results on ripples and prefrontal units, UP and DOWN state modulation was not affected by olfactory deafferentation, suggesting that RCD is sufficient to organize the cortical dynamics (Fig. 6m,n).”

We further discuss this topic in the discussion:

“Here we demonstrated that the respiratory rhythm coordinates cortico-hippocampal dynamics as well as unit activity across the limbic system, with RCD being a sufficient mechanism. . . . Understanding the causal role of respiratory entrainment in the synchronization of cortico-hippocampal dynamics will require dissection of circuit mechanisms of RCD and fine-timescale, closed-loop optogenetic perturbations.”

In Fig. 8, the UP and Down mark the state onset, but it is not clear where the start is. They do not seem to coincide with higher and lower firing rates in the cortex.

We thank the reviewer for the suggestion. We have now added arrows to demarcate the onset of the UP and DOWN periods. We have also slightly modified the schematic to better depict the relationship of these states with the cortical firing rate.

Reviewer #2

Summary

The authors found that localized neural delta oscillations and gamma bursts in prefrontal cortex, nucleus accumbens and basolateral nucleus of the amygdala as well as hippocampal ripples are locked to breathing rhythm. Furthermore, they showed that spiking activity of prefrontal cortex, visual cortex, midline and sensory thalamus, nucleus accumbens and basolateral nucleus are phase-locked to breathing cycles. Next, they showed that after olfactory deafferentiation the locking of neuro-oscillations in prefrontal cortex to the breathing cycles is interrupted and spiking activity is more disconnected from breathing. Lastly, using optogenetics, they show that spiking activity in the nucleus accumbens and prefrontal cortex is modulated by both ripples and breathing cycle.

Presentation and Structure

The overall structure of the data presentation is well done, and the overall concept, experimental strategy and discussion are somewhat easy for the reader to follow. A potential point of improvement would be to improve the description of experimental set ups for each procedure (or figures), improve the arrangement of figures and graphs, even though I do understand that the study contains both a descriptive and an experimental part with some inconsistencies and unavoidable overlaps. Lastly, the outlook addresses some general concepts in a manner that is quite superficial and overambitious where it is difficult to understand what the authors' take on their finding and their message to the field is, as for instance, a more detailed look into specific behaviours controlled by medial prefrontal cortex (see below).

We thank the reviewer for the helpful suggestions towards improving our manuscript.

We address the specific points below.

Major points

- **Explanations of the experimental setups are at times not very clear:**

We thank the reviewer for noting this weakness in our manuscript. In the revised version we strived to improve the description of the experimental setups throughout the text. We explicitly stated the configuration used in each results and legend section and we included schematics of the procedures in the new Supplementary Fig. 9. We include more details about the changes and additions below.

- **How was "quiescence" defined? Did the authors differentiate between quiet wakefulness and sleep (with NREM and REM)?**

Here, we defined quiescence and sleep periods based on the arousal state of the mouse, as defined based on the camera captured micromotions and used the cortical and hippocampal LFP power to further differentiate states (described in Supplementary Fig. 9).

More generally, there is a growing consensus in the field that there is a continuum of brain states from active wakefulness to quiescence and through drowsiness to sleep. Although it is in principle possible to segment these brain states based on arbitrarily thresholds of various parameters (e.g. LFP power), the described patterns of activity, such as gamma oscillations, ripples, and UP/DOWN states occur with varying probability throughout quiescence and sleep. Further, accurately defining the transition between quiescence and sleep even in freely-moving animals, and even more so in the head-fixed configuration is a challenging goal. For these reasons, here we opted to conceptually pool the results from the quiescence and sleep states (with the exception of Fig. 1) in order to avoid the bias of the arbitrary segmentation and importantly, to highlight the generality of the described overarching phenomenon of the respiratory modulation of neuronal activity and network dynamics.

- **Do Fig. 2-4 all belong to data from the same mice? Can you briefly introduce the new experiment after the one for Fig.1?**

We have now included the following statement in the results section, in order to introduce the experiments in Fig. 2-4.

“To further investigate the extent of respiratory entrainment of prefrontal circuits at the level of neural population activity, we examined the firing of extracellularly recorded single neurons in mPFC in relation to the respiratory phase (Fig. 2a). To achieve that, in addition to the wire-electrode recordings in freely-behaving mice, we performed large-scale silicon probe recordings from head-fixed mice (Supplementary Fig. 9). This configuration enables the effective recording from a large neuronal population and across multiple experimental conditions.”

In addition, we have performed a number of changes in the text, described in response to the other points.

- **Why were some mice head-fixed others not for Fig. 2-4?**

For a subset of the analyses performed (such as the phase modulation of neurons and gamma bursts), it is possible to use data from recordings in head-fixed mice using silicon probes and in freely-behaving mice using single-wire electrodes. In these cases, we pooled the data,

highlighting the generality of the described phenomena. In the rest of these analyses though, we have used data only from large-scale silicon-probe recordings, since these are necessary for the anatomical characterization of the phenomena (within prefrontal regions and across multiple brain regions). For each panel, we have clarified the configuration used for each recording.

Could the authors explain explicate the experimental setups more clearly in the text and "walk the reader through the process" in more details? Could the authors also provide simple schematics in each figures where a new experimental procedure (or animal preparation, e.g., tetrode vs (multi-)shank recordings) is introduced that show the whole setup in terms of where electrodes are? It appears important to also clarify whether if the mice are freely moving head-fixed for each result sections. Accordingly, the authors should explain why some mice were head-fixed, others not.

We agree with the reviewer that the number and complexity of the described experiments can make it challenging to follow the thread. For this, we have added further explanations throughout the text to make it easier to follow the results and added schematics of these procedures (Supplementary Fig. 9a-h), to which we refer in the relevant results and methods sections. We have specified the head-fixed or freely-behaving configuration in all the results sections and legends. Finally, we added a paragraph (see below) explicitly explaining the choice of head-fixed recordings for part of the experiments.

"To further investigate the extent of respiratory entrainment of prefrontal circuits at the level of neural population activity, we examined the firing of extracellularly recorded single neurons in mPFC in relation to the respiratory phase (Fig. 2a). To achieve that, in addition to the wire-electrode recordings in freely-behaving mice, we performed large-scale silicon probe recordings from head-fixed mice (Supplementary Fig. 9). This configuration enables the effective recording from a large neuronal population and across multiple experimental conditions."

Could the authors also explain how behaviour was scored, how the different stages were differentiated from each other and give some exemplary graphs and figures of the process?

We thank the reviewer for this suggestion.

We have included in Supplementary Figure 9e-f, example figures of the approach and the process used for the behavior classification. Since description of the procedure graphically is

not efficient at capturing all the technical details we have further expanded the results and methods section, in order to better describe this procedure.

Results section

“Using the density of head micromotions and muscle twitches, we classified behavioral segments as active awakening, quiescence, or sleep (Supplementary Fig. 9i,j) (see Methods). Distinct states are associated with changes in the breathing frequency (Fig. 1c).

...

To achieve that, in addition to the wire-electrode recordings in freely-behaving mice, we performed large-scale silicon probe recordings from head-fixed mice (Supplementary Fig. 9c-d). This configuration enables the effective recording from a large neuronal population and across multiple experimental conditions. For the detection of quiescence and sleep periods for head-fixed recordings, we relied on video tracking of the mouse snout and body, from which we derived a micromotion signal, similar to the freely-behaving case.”

Methods section

“Additionally, the activity of mice was tracked using an overhead camera (Logitech C920 HD Pro). The camera data were transferred to a computer dedicated to the behavior tracking and

were acquired and processed in real-time using a custom-designed pipeline based on the Bonsai software (Lopes et al. 2015). Video data were synchronized with the electrophysiological data using network events. Video was preprocessed to extract the frame-to-frame difference and calculate a compound measure that we found provided an excellent proxy for the behavioral state. Video frames were thresholded and binarized (Supplementary Fig. 9e-h). A logical exclusive OR operation was applied on consecutive frames, a calculation that provides the effective frame difference. The sum of these differences provides a measure of overall change between consecutive frames. We found that the changes in the amplitude of variance of this measure over time are informative for the current state of the mouse. Complete immobility is easily distinguishable using this measure, due to the low amplitude and small variance of the signal. A threshold was set manually such that even small muscle twitches during sleep were captured, but breathing-related head-motion was below threshold. Using the density of head micromotions and muscle twitches, we were able to classify behavioral segments as active awakening, quiescence, or sleep (Supplementary Fig. 9i,j). For head-fixed recordings, we relied solely on high-resolution video of the mouse snout and body, from which we derived a micromotion signal that was used in the same way as the jerk-based signal for freely-behaving mice.“

- **Lines 200-201, 251 of the main text: the authors mention gamma rhythms in the PFC are generated through input from the olfactory bulb, but they only show the phase-amplitude coupling of gamma to respiratory cycles (both in Fig.4 and Fig.5). However, it might also be possible that gamma bursts are regulated through the delta frequency in PFC and this establishes the link between gamma bursts and breathing cycles. Consequently, the lower modulation index could be due to the decoupling of delta and breathing rhythm. Could the authors provide additional gamma burst analyses (i.e. power, duration and occurrence rate of gamma bursts before and after deafferentiation) to support their claim?**

We thank the reviewer for pointing out this potential interpretation of the results.

Following OD, it is the general power of gamma activity that is reduced, not only its modulation by breathing/slow oscillations. As suggested, we have now included further analyses, showing the reduction in the gamma power and the reduction in the coordination of mPFC gamma bursts with OB gamma bursts (Fig. 5g, Supplementary Fig. 6h). We have now rephrased this sentence to reflect more precisely the observed effect.

“This manipulation in both freely-behaving and head-fixed mice eliminated the respiration-coherent and spectrally-narrow prefrontal slow oscillatory LFP component (Fig. 5a-d, Supplementary Fig. 7g-h), consistent with the disappearance of the CSD sink in deep layers (Fig. 5r), while at the same time significantly reduced the power and correlation of prefrontal gamma oscillations to those in the olfactory bulb and abolished their entrainment by the breathing rhythm (Fig. 5f,g, Supplementary Fig. 6h), without altering the respiratory dynamics (Supplementary Fig. 7b-r).”

- The chosen frequency of the optogenetically induced ripples is comparatively low (70-250 Hz) for mice (a frequency range from 100Hz-250Hz is usually found in the literature. Ripples in rats are often described to have a lower range). Indeed, the authors themselves show that the spontaneous ripples they detected have a frequency range around 150Hz. Could the authors explain why they chose a lower frequency range? And could they provide some descriptive analyses of their induced ripples, spontaneous ripples and perhaps high gamma bursts (i.e. average power, peak frequency, duration, number of cycles, symmetry of the event)? That would strengthen the authors' optogenetic experiments and support their claims.

We thank the reviewer for these suggestions.

In these experiments, the optogenetic induction of the ripples was performed by stimulating the dCA1 region using a single smooth half-sine wave (200ms duration; 1-3s ITI). As a result of this stimulation, the local hippocampal circuit oscillates in a ripple-like fashion following ING/PING mechanism underlying endogenous ripples (Stark et al., Neuron, 2014). The frequency of the generated ripples is thus not imposed externally, but it is rather generated natively in the circuit. This somewhat lower frequency of the optogenetically induced ripples agrees with other reports in the literature (Stark et al., Neuron, 2014; Fernández-Ruiz et al., Science, 2019), though the exact reasons for this are not yet understood. Based on theoretical work (Geisler et al., J. Neurophysiology 2005) the exact frequency of the oscillation depends on the balance of the drive of excitatory and inhibitory populations, which clearly is different between endogenous sharp-wave-associated Schaffer-collaterals input, that targets both pyramidal cells and interneurons, and optogenetic excitation that targets pyramidal cells due to *CaMKIIa*-driven opsin expression. However, these artificial ripples appear to be qualitatively and functionally related to the intrinsic ripples, given their capacity to enhance behavior and learning and to extend the neuronal ensemble content of intrinsic events (Stark et al., PNAS 2015; Fernández-Ruiz et al., Science, 2019).

Following the advice of the reviewer, we performed a further characterization of the properties of the induced ripples (Supplementary Fig. 8), including the evoked response of dCA1 units

following the optogenetic stimulation (Supplementary Fig. 8f) and the distribution of frequency, power, and number of cycles of the optogenetically ripples (Supplementary Fig. 8g-h).

Direct comparison of these parameters, based on the points above, does not, in our view, affect interpretation of the results. While the average endogenous ripple contains 9 cycles, the average opto-ripple contains twice as many, and, given that it is twice slower, lasts 3-4 times longer. Likewise the power of opto and endogenous ripples, as well as synchrony across the hippocampus cannot be directly compared. Yet, qualitatively and, importantly mechanistically, optogenetically-induced ripples activate the same population dynamics in the CA1 and, given behavioral effects cited above, similar activation of the downstream targets. Thus, we believe that using opto-ripples for probing the excitability modulation provided by the respiration cycle is appropriate.

- **About general aspects of the experiment inducing ripples with optogenetics: Could the authors explain the rationale of the experiment and clarify the interpretation of the results? Right now, it is not clear in the manuscript what the authors try to show with this experiment: they induce ripples and lock them to breathing cycle and they see similar activity in PFC as in spontaneous ripples. However, how does this replication of an observed pattern show that it really is breathing that causally modulates coupling of hippocampus and prefrontal cortex? What would we see in the experiment if breathing was not the synchronizing cause? If possible, could the authors redo the analysis in Fig.7s with the data of the deafferentiation experiment? This would actually support what they want to claim with the optogenetics experiment.**

We thank the reviewer for the careful reading and insightful discussion about this experiment.

Based on the previous results in this work, we have established that ripples occurring in distinct phases of the breathing cycle differentially affect the downstream targets. However, it is unclear whether this is due to some difference in the “content”/neuron-participation in the ripple, co-occurring inputs from other regions to the mPFC or spontaneous co-fluctuations of the activity. To clarify these potential mechanisms, we performed the optogenetic experiment.

The optogenetic manipulation establishes two facts:

- a. That the evoked activity in the mPFC is directly linked to the dCA1 ripple activity.
- b. That the respiratory modulation of the magnitude of the endogenous ripple-evoked activity is due to the differential responsivity of the mPFC in distinct phases of breathing.

This clarifies that the breathing modulation of prefrontal excitability underlies the differential coupling with the hippocampus across phases. This mechanism might be more general, given that evoked responses in V1 by the visual stimuli followed a qualitatively similar respiration-modulated excitability profile (Supplementary Fig. 8n).

However, this finding is agnostic to the underlying mechanism of the mPFC modulation (through the olfactory reafferent ROR, the efference copy RCD or any other).

Indeed, the reviewer is absolutely right that this analysis provides an opportunity to investigate the mechanism underlying the modulation of prefrontal excitability. So, as suggested, we performed the same analysis for intrinsic ripples following olfactory deafferentation (updated Fig. 7s).

In agreement with the previous results on the relationship of ripples with UP and DOWN states, it appears that the RCD is the primary modulator of prefrontal excitability, since OD leaves the respiratory modulation of the prefrontal excitability largely intact.

We added respectively in the Results:

“... The persistence of the excitability modulation of mPFC following OD (Fig. 8s), further reinforces the notion that RCD is the main mechanism behind this phenomenon.”

And in Discussion :

“... Importantly, opto-ripple-based experiments demonstrate that it is the breathing cycle and not other covariates of endogenous ripples that modulates cortical excitability, while OD experiments suggest that this phenomenon is mediated by RCD.”

- **About the outlook in general: The statement about the default mode network seems a bit out of place. Even though the authors showed that breathing influences several remote brain regions, I would still be hesitant to link it to a quite specific overall brain state, especially since it is rather mentioned in the context of MRI measurements and less so in the electrophysiological field. At the same time, since breathing is not only linked to LFP and spiking as described but to several physiological features of the entire organism (i.e. blood pH, emotional state, cardiovascular state via brain stem influence) such a statement could also be dismissed as rather generic as chances are high that**

breathing could have an indirect influence on the default mode network via effects on other organs.

We agree with the reviewer about the complexity of identifying the relationship between breathing entrained networks and the default mode network as defined in the context of fMRI studies. To avoid the misinterpretation of these statements in the outlook, we have modified this paragraph, as below, and avoided referring to the DMN but rather to the more relevant concept of neuronal assemblies and circuits.

“Finally, in light of the wide modulation of multiple circuits by breathing during quiescence, we suggest that the entrainment by breathing potentially defines functional sub-networks within and across neural circuits. To examine this hypothesis future work will be needed to carefully examine the fine temporal structure of neuronal assemblies and their modulation by the RCD and ROR copies of the breathing rhythm throughout cortical and subcortical structures, an endeavor that might uncover such functional sub-networks involved in distinct behaviors.”

In the beginning of the study, the authors looked at breathing during fear. It would be much more compelling if the authors gave thoughts on possible implications of breathings influence on medial PFC function during fear behaviour as this brain region has been described extensively in this context. They could transfer the paragraph form lines 402-411 to the outlook.

We thank the reviewer for this suggestion. We followed this advice and extended this section to discuss the implication of the mPFC entrainment by breathing for fear behavior. We include both the previous and new paragraphs under the new heading “*Respiratory entrainment - a potential substrate of fear behavior*”, as below.

“*Respiratory entrainment - a potential substrate of fear behavior*”

Extending the generality of respiratory rhythm entrainment, we show that fear-related prefrontal 4 Hz oscillations (Karalis et al. 2016, Dejean et al. 2016) are a state-specific expression of this entrainment and originate from the refferent respiratory entrainment of olfactory sensory neurons by passive airflow (Grosmaître2007) (Supplementary Fig. 2).

Fear memory and expression relies on the coordinated activity in the mPFC and BLA. The joint respiratory modulation of prefrontal and amygdala circuits might contribute to this coordination during fear behaviour. The pronounced LFP amplitude and narrow respiratory and LFP frequency during this state (Supplementary Fig. 2) is potentially fine tuning the temporal coincidence of ensemble activity in these regions, resulting in increased coherence and co-firing (Karalis et al. 2016).

Importantly, although prefrontal 4 Hz LFP oscillations originate in fear-associated enhanced breathing, the ROR is not necessary for the expression of innate or conditioned fear behavior (Supplementary Fig. 2), in agreement with a recent report (Moberly et al. 2018). This suggests

the independence of fear expression from the respiratory entrainment or the potential sufficiency of RCD for the expression of fear behavior.

Consistent with such mechanism, the optogenetic induction of 4 Hz oscillations in prefrontal circuits is sufficient to drive fear behavior in naïve animals (Karalis et al. 2016), raising the possibility that this effect is mediated by the bidirectional interaction of prefrontal networks with the respiratory centers via top-down projections to periaqueductal gray and the ascending feedback via RCD giving rise to system-level resonance at breathing frequency. This sets the stage for future investigations of the interaction between the RCD and ROR in limbic networks and, in turn, the top-down modulation of breathing and emotional responses.“

Minor points

- **Fig. 2g: Could the authors explain how the power was normalized?**

In this plot, the LFP power for each electrode across in the dorsoventral axis was calculated and the power profile was z-scored (across depth) to quantify the relative change of the signal for each animal. We added a clarification about this point in the legend of the figure.

- **Fig. 2i-j: What does time point zero exactly refer to?**

In this analysis, we are shifting the spikes of each neuron in relation to the ongoing phase of the respiratory signal. This shift enables us to identify lags that improve the phase modulation of the neurons, due to the non-uniformity of the phase of the signal. Time zero corresponds to the original, non-shifted data. We have now clarified this point in the legend of the figure.

- **Fig. 2h: Could the authors mention that it is the spike-phase modulation that is shown here?**

We clarified this point in the legend of the figure.

- **Fig. 4b: Could the authors address the fact PFC shows a greater phase-amplitude coupling than OB? How could that be? Please comment and discuss this finding.**

This is likely due to the fact that for these analyses the OB gamma is calculated from the LFP of a superficial electrode on the skull, in contrast to the higher SNR of mPFC gamma recorded using the silicon probes in depth. We now discuss this point in the legend of the figure.

“The difference of the modulation magnitude between the two signals reflects the higher SNR of the silicon probe recordings in mPFC compared to the superficial electrode above the OB.”

- **Title of Fig.5: Title says that there is no neural entrainment, however 5h-5j seem to show that it is the case and the main text also states that it does. Please clarify.**

The title of Fig. 5 was meant to highlight the fact that after OD there still remains a large fraction of neurons entrained by breathing (via the RCD pathway). However, we agree with the reviewer that this phrasing can be confusing, so in the revised version of the manuscript we rephrased this title to: *“Reafferent respiratory input underlies respiratory LFP entrainment.”*

- **Fig. 7s: Please clarify how PFC activity was normalized.**

We clarified this point in the legend of the figure.

- **Fig.7: Could the authors provide the full name of the viral construct used in the main text and a simple schematic in the figure to support the reader?**

We added the full name of the construct in the main text, in addition to the methods. We also modified the schematic in Fig. 7k to explain the injection of the excitatory opsin and the implantation of the optic fiber in dCA1 and the silicon probe recording in the mPFC.

We also briefly described the procedure in the legend of this panel.

“(k) Schematic of the experimental design for testing the effect of optogenetically induced hippocampal ripples on the prefrontal network. An excitatory opsin (AAV2/9-CaMKIIa-ChETA) was injected in the dCA1 and a silicon probe paired with an optic fiber was inserted to generate opto-rippled and to record the induced activity. Simultaneously, a silicon probe was implanted in the mPFC to record the effect of the intrinsic and optogenetically induced ripples.”

- **Supp. Fig.7: Could the authors also show anatomical pictures of olfactory bulb and prefrontal cortex to show that these regions were not affected by the injections?**

We have included example histological reconstruction of the olfactory bulb, prefrontal cortex, and hippocampus from methimazole injected mice, next to images from control (saline-injected) mice (Supplementary Fig. 7n-p). No gross anatomical differences were observed between the two groups.

- **Supp. Fig. 1:**
- **Please briefly describe what a Thermistor is.**

We have now included a description of the thermistor in the legend of the figure.

“Thermistors are temperature-dependent resistors. When placed in or near the nasal cavity, they report the change in the temperature of the incident air flow due to breathing, since warmer air is exhaled and colder air inhaled (McAfee et al. 2016).”

- **Panel i: Could the other briefly mention what SNR stands for and give a short interpretation of the panel?**

We have now included in the legend of the figure (now Supplementary Figure 1g) a clarification of the SNR and an interpretation of the analysis in this panel.

“Time-reversal control for the effect of signal-to-noise (SNR) differences on Granger causality estimates. If the effect of GC is mediated by the higher SNR of one signal compared to the other, reversing the signals in time should fully reverse the directionality of the calculated causality. In contrast, we observe that the calculated GC is higher for the forward (original) direction, compared to the reversed signals, suggesting that this is due to the underlying Granger causal relationship between the two signals.”

- **EOG is commonly understood in the field as "electro-oculogram". Please change the abbreviation (e.g. Olfactory EEG)?**

We thank the reviewer for this suggestion. We have now followed this advice and renamed EOG to Olfactory EEG.

- **lines 125-134 of main text: the authors explain the phase-modulation with poly-synaptic projections. Since the modulation follows a dorso-ventral gradient, is it possible that there is after all a possible mechanical influence due to the air-flow through the nasal cavity which is situated right below this part of the brain? Please address this point in the discussion.**

We thank the reviewer for highlighting this interesting point and for inspiring the following analysis. To investigate this possibility we performed the same analysis (now included in Supplementary Fig. 3m) following the OD manipulation. This analysis shows that the DV gradient of the LFP power is reduced following OD, supporting the notion that the poly-synaptic pathway associated with the olfactory-related inputs to the mPFC subregions is responsible for the generation of the DV gradient. Given that the unit modulation across mPFC depth and across the various regions investigated doesn't correlate with the dorsoventral location of the region.

In parallel, we have performed a number of analyses to establish that the unit modulation and waveshape do not correlate with its distance from the silicon probe, which would be expected if

the source of the modulation was due to the pulsation of the brain (Supplementary Fig. 3i-k). These evidence suggest that passive pulsation due to airflow is an unlikely source for these phenomena.

We further discuss this point in the discussion.

“Anatomically-resolved analysis of prefrontal OB-generated current sources and unit activity suggests that deep layers and mostly ventral regions are the main targets of OB refference and give rise to observed LFP signals. The reduction of the dorsoventral LFP profile following OD is consistent with olfactory related sources of this gradient, rather than brain-motion related activity due to airflow through the nasal cavity. Although the interpretation of these gradients is challenging given the existence of volume conduction from the OB, these findings suggest a potential functional role of the differential modulation of orbital, prefrontal, and cingulate regions and is worthy of future investigation.”

- **Lines 145-176 of main text:** in the last part of the paragraph the entorhinal input to dentate gyrus is mentioned. Could the authors briefly explain why those inputs are important here? Please cite some helpful sources for non-expert readers who don't know the specific anatomy of the hippocampal formation. Please discuss these possible connections.

We thank the reviewer for this suggestion, which indeed improves the readability of the manuscript. We have now expanded that paragraph, to include the following information:

“Inspiration was associated with an early sink in the outer molecular layer of DG, indicative of input from the layer II (LII) of the lateral entorhinal cortex (LEC), followed by a sink in the middle molecular layer of DG, indicative of input from the layer II of the medial entorhinal cortex (MEC) (Bragin et al. 1995, Buzsaki et al. 2003) (Fig. 3d,e, Supplementary Fig. 5e). The inputs from MEC and LEC are the two primary cortical inputs to the hippocampus, providing primarily spatial and sensory information respectively (Hargreaves et al. 2005, Henriksen et al. 2010, Fernandez-Ruiz et al. 2019). The modulation of these inputs by breathing reinforces the notion that the hippocampal respiratory modulation is mediated by entorhinal inputs, while the distinct

timing of these two inputs suggests that the respiratory phase might serve as a reference for the temporal organization of the incoming information.

- **Lines 256-258 of main text: Could the authors cite some sources at this point and make clear that this circuit has already been proposed? It comes across as if the authors came up with the term.**

This phenomenon of the non-olfactory entrainment of cortical neurons by respiration via a potential intracerebral, centrifugal efference copy mechanism is novel and has not been described in the literature. We thus coined the term “respiratory corollary discharge” (RCD) to describe this unknown to-date phenomenon by analogy to corollary discharge phenomenon that is associated with motor-event-triggered efference copy.

- **Lines 277 of main text: the authors mention an observed current sink in medial entorhinal cortex in sup. Fig. 8c, d but the description of the figure says that is was recorded in dorsal Hippocampus.**

This sentence refers to the CSD sink in the dorsal hippocampus due to the MEC LII input. However, the wording is confusing, so we have rephrased as follows:

“Keeping up with the role of the entorhinal input in mediating respiratory drive on ripples, we observed a consistent relationship between the magnitude of the current sink in DG mol. layer directly preceding ripple occurrence and the phase within the respiratory cycle of the ripple occurrence (Supplementary Fig. 8c,d), suggesting that ripples occurring at the preferred phase of respiration follow a strong MEC LII input.”

- **Lines 299-300 of main text: Can the authors cite sources for the claim of "RCD-mediated inputs to DG"?**

This phrase refers to the inputs (CSD sinks) identified in the current study (Fig. 6c). This was not clear because of the missing reference and description of this figure. We have now rephrased, as below, this section to make it clear that this is a fact identified in the current study.

“This is consistent with the respiration-related synaptic input to the DG middle molecular layer preceding ripple events (Fig. 6c), which persists following OD (Supplementary Fig. 8b), which suggests an RCD-mediated coordination of SWR occurrence with the cortical UP states known to be mediated via the MEC (Isomura et al.,2006) (Supplementary Fig. 8c,d).”

- **Lines 379-385 of main text: Could the authors mention some sources for interested readers?**

We thank the reviewer for noticing the lack of citations in this paragraph. We have now updated this paragraph with references to the relevant literature.

“Brainstem circuits are well-known to generate breathing rhythm (Del Negro et al. 2018, Cui et al. 2016), and at the same time send massive diffuse ascending projections to the forebrain (Yackle et al. 2017, Yang et al. 2018), thus the most parsimonious explanation of the observed phenomenon is a directional drive from the brainstem to the forebrain. While descending feedback inputs (Yang et al. 2020), might modulate breathing and contribute to the phenomenon, they are not serving the rhythm generation function, which is known to be implemented in the brainstem circuits that provide a causal effect on the forebrain circuits (Del Negro et al. 2018). OD experiments result in a virtually absent spectral peak in the prefrontal LFP and reduction of gamma and unit entrainment across different circuits (Fig. 5), while respiratory activity is unchanged. This strongly suggests the same directionality of the phenomenon as suggested by the analytical methods.”

“It is likely, however, that via descending cortical projections (Yang et al. 2020), cortical SO provides feedback to the pontine respiratory rhythm-generating centers and thus the interaction between respiratory dynamics and slow oscillations could be bidirectional.”

Reviewer #3

In this article, the authors record from several brain areas of the mouse brain, both local field potential and single-unit activity, to gather evidence related to the hypothesis that breathing serves as a brain-scale coordinator of neural activity. They center their efforts in the so-called limbic system but record from other areas as well. Also, the work focuses on what the authors call "off-line" behavioral states, i.e., periods of inactivity and sleep.

The authors found widespread respiratory modulation of LFPs and single-unit activity across brain areas, consistent with previous findings. They also characterize the incoming synaptic volleys associated with respiratory modulation through current source density analysis. In addition, they studied the role of this modulation in hippocampal-cortical communication and the respiratory modulation of cortical UP/DOWN states. Finally, to address the origin of the modulation, they intervened animals by eliminating the input to the brain from the olfactory epithelium (i.e., deafferentation), which is mechanically stimulated in each respiratory cycle and relays this activity to the olfactory bulb. The deafferentation effectively abolished most LFP and a fraction of the single-unit modulation. However, some single-unit respiratory modulation survives OB deafferentation, strongly suggesting a corollary discharge from the respiratory brainstem.

The work is well-executed, the evidence is compelling, and the terrain covered is vast. Several simultaneous and different sources of evidence are provided, including behavior, electrophysiological recordings, anatomical/histological, and interventions (pharmacology, optogenetics). Evidence of this quality is relevant for the field and can contribute to understand some of the big-picture puzzles in neuroscience.

We thank the reviewer for positive evaluation of our work.

I do have, however, some observations, comments, and requests for the authors.

1) There is a lack of clarity regarding the actual behavioral states or periods used to get the electrophysiological data. For example, it was not always clear when the data came from quiescent periods, when from sleep, or if quiescent and sleep periods were pooled together. Are both states equally represented in the data? Except for one or two plots, this is not clear.

A diagram or plot containing the relative amounts of data from each period should be provided separately for each animal. I'd also include here labeling of the head-fixed data.

1.1) Also, what is the justification for pooling (at least conceptually) data from quiescent periods and sleep?

There is a growing consensus in the field that there is a continuum of brain states from active wakefulness to quiescence and through drowsiness to sleep. Although it is in principle possible to segment these brain states based on arbitrarily thresholds of various parameters (e.g. LFP power), the described patterns of activity, such as gamma oscillations, ripples, and UP/DOWN states occur with varying probability throughout quiescence and sleep. Further, accurately defining the transition between quiescence and sleep even in freely-moving animals, and even more so in the head-fixed configuration is a challenging goal. For these reasons, here we opted to conceptually pool the results from the quiescence and sleep states (with the exception of Fig. 1) in order to avoid the bias of the arbitrary segmentation and importantly, to highlight the generality of the described overarching phenomenon of the respiratory modulation of neuronal activity and network dynamics.

For this reason, we defined quiescence and sleep periods based on the arousal state of the mouse, as defined based on the camera captured micromotions (as described in the new Supplementary Fig. 9) and used the cortical and hippocampal LFP power to further differentiate states.

For the analyses pertaining to specific network patterns of activity (ripples, UP/DOWN etc) we independently detected them during behaviorally defined immobility periods. For the analyses of neuronal spiking activity in relation to the respiratory phase, we utilized all periods of behavioral immobility.

We have clarified these points in the Results and Methods section. Following the advice of the reviewer, we also include in Supplementary Fig. 9k a visualization of the relative periods of time for included animals suggesting that these two states contributed approximately equally to the results.

“Using the density of head micromotions and muscle twitches, we classified behavioral segments as active awakening, quiescence, or sleep (Supplementary Fig. 9i,j) (see Methods). Distinct states are associated with changes in the breathing frequency (Fig. 1c).

...

To achieve that, in addition to the wire-electrode recordings in freely-behaving mice, we performed large-scale silicon probe recordings from head-fixed mice (Supplementary Fig. 9c-d). This configuration enables the effective recording from a large neuronal population and across multiple experimental conditions. For the detection of quiescence and sleep periods for

head-fixed recordings, we relied on video tracking of the mouse snout and body, from which we derived a micromotion signal, similar to the freely-behaving case.

...

Additionally, the activity of mice was tracked using an overhead camera (Logitech C920 HD Pro). The camera data were transferred to a computer dedicated to the behavior tracking and were acquired and processed in real-time using a custom-designed pipeline based on the Bonsai software (Lopes et al. , 2015). Video data were synchronized with the electrophysiological data using network events. Video was preprocessed to extract the frame-to-frame difference and calculate a compound measure that we found provided an excellent proxy for the behavioral state. Video frames were thresholded and binarized. A logical exclusive OR operation was applied on consecutive frames, a calculation that provides the effective frame difference. The sum of these differences provides a measure of overall change between consecutive frames. We found that the changes in the amplitude of variance of this measure over time are informative for the current state of the mouse. Complete immobility is easily distinguishable using this measure, due to the low amplitude and small variance of the signal. A threshold was set manually such that even small muscle twitches during sleep were captured, but breathing-related head-motion was below threshold. Using the density of head micromotions and muscle twitches, we were able to classify behavioral segments as active awakening, quiescence, or sleep (Supplementary Fig. 9i,j). For head-fixed recordings, we relied solely on high-resolution video of the mouse snout and body, from which we derived a micromotion signal that was used in the same way as the jerk-based signal for freely-behaving mice. “

1.2) I'm also curious, why call these states "offline" without quotation marks? Is this a technical term in use in the sleep research field? I'd explain a bit about this.

Indeed, offline is a term commonly used in relation to sleep and refers to the sensory disconnection of cortical circuits. Here, we use it to refer to quiescence and sleep states, as opposed to active (exploratory) wakefulness. We followed the advice of the reviewer and clarified this in the introduction.

“During offline brain states (such as sleep), the cortex is sensory disconnected from the environment. Systems consolidation across distributed circuits has to rely on the global coupling of internal network dynamics, to enable the coordinated reactivation of previous experiences across remote brain regions.”

2) I would like to see a detailed overview of the numbers of neurons modulated and non-modulated, per area and animal. I'm thinking of a table, or figure/diagram, containing the brain areas as columns and individual animal subjects as rows, with the number and percentage of modulated neurons as entries. Getting a good sense of the variability/heterogeneity of the responses, both across areas and especially across animals, is as informative of the phenomenon as the "population average" message. I see the inter-subject variability as crucial data, not as noise. In our times of replicability

problems and of open science approaches, data like these are a must. If space is an issue, I think this is much more informative for researchers than example traces and/or anatomical diagrams.

We agree with the reviewer's statement.

We have now included this data in Supplementary Fig. 9l.

3) I think all the data presented in boxplot format, given that it comes from small samples (e.g., n=6, n=7), should instead be provided as individual jittered data points, plus a bar indicating the mean or median. For both supplementary and paper figures. In methodologically complex studies as this, I understand small samples. Precisely because of this, I think data should be more visible to get a better sense of the evidence gathered.

We agree with the reviewer on this remark. We have updated all bar and box plots to include the individual data points, including the median and 25th and 75th percentiles.

4) Can you provide more information about the methimazole treatment?

- the general state of the animals after treatment (ie, sleep vs awake profiles, weight time series, feeding amounts, etc)?
- a more long-term respiratory data after treatment?

Given that it is not a widely used intervention, at least in neuroscience, and that it is a systemic perturbation, it would be useful to get a sense of the general impact of this intervention.

We thank the reviewer for prompting us to perform this characterization.

We have now included in Supplementary Fig. 7m, the characterization of the activity profile of mice before and after the methimazole treatment, showing stable circadian rhythms in the activity of the mice following OD.

In addition, in Supplementary Fig. 7l we plot the weight of the mice on the days before and after the treatment. No significant change in the weight of the mice was observed following the treatment.

In Supplementary Fig. 7d,e we calculate the power spectrum of the respiratory signals of the mice before and after the treatment, and finally in Supplementary Fig. 7n-p we provide histological reconstruction of the OB, mPFC, and hippocampus of mice following the methimazole treatment, in response to the request by reviewer 2, showing no gross anatomical changes.

5) I think statements like "we identified an intracerebral centrifugal respiratory corollary discharge" or "identified a novel global mechanism" may be a bit of an overstatement. The finding is clear, and there are not too many options for the origin of the signal after deafferentation, but the data has not properly identified the pathway.

We agree with the reviewer. We have now removed or rephased such statements where they occurred.

COMMENTS ABOUT FIGURES AND OTHER POINTS:

FIG 1

-1a It would benefit from showing an inhalation-triggered area LFP mean, compared with a null model constructed by taking the trigger from random positions in the time series.

We thank the reviewer for this suggestion. We have now included this analysis in Supplementary Fig. 1i.

Note: we indicate the error with std rather than sem, to make the error bars visible, which would not be the case with sem, due to the large number of events.

- 1b. Top signal ("Motion"). The caption doesn't refer to it.

We have now included a description of this signal in the figure caption.

- 1b. The spectrogram shows a peak of activity right at 4Hz. But in Fig 1c 1d & 1e peak is at 3 Hz. Why is this?

- I'd include some variability metric for all the spectra (example g, i).

In Fig 1c we characterize the distribution of the peak frequencies across states. As can also be seen in the example spectrogram in Fig 1b, as the state of the animal changes, the respiratory frequency changes continuously. It tends to be centered at ~3Hz during long quiescence periods.

We have included the quantification of the peak frequency for all animals in Supplementary Fig. 1h.

We have also included in Supplementary Fig. 1e,f the variability metric for the spectra of coherence and Granger causality across states.

FIG 2

- 2a: Mean responses need a variability metric.

We have now included the more detailed inhalation-triggered LFP analysis in Supplementary Fig. 1i, as suggested above.

We opted for leaving this simpler single line plot here to serve as a schematic of the phase.

- 2c: legend says "horizontal line"; should be "vertical"

We have now replaced this.

Many anatomical structures lack full naming (e.g., MO, VO, IL, DP, etc.)

We have now included the abbreviation for all brain regions.

- 2k: a cross indicating axes (ap, dv) would be of help here.

We have now included the axes in this panel.

k

FIG 3

-3d: I'd change the color of the star and circle, as they are hard to see.

We have followed the advice of the reviewer and replaced the color of the star and circle with white.

FIG 5

-5l: for completeness, I'd add the MEC LII sink comparison as well.

We thank the reviewer for this suggestion. We have now replaced this panel with the ratio of the MEC and LEC sinks.

FIG 6

6f: Letter F is uppercase

We have changed the capitalization of the letter.

6m,n: why including the histogram example only for n and not for m?

We have now included the example histogram in panel m as well.

6o,q: why/how is probability expressed as stdev in these plots?

In these plots, the ripple occurrence histogram before/after the onset of each is normalized for each mouse and plotted. We agree that the term probability is confusing here, so we have replaced it with occurrence rate and we clarify this in the legend of the figure.

FIG7

7a,m What information exactly do we get by looking at the real part of the transform?

Both phase and magnitude of the signal contribute to the real part of the wavelet transform. This compound measure highlights the phase-magnitude relationship to the ripple peak centered to the largest trough and provides a hybrid view between the raw signal and the spectral decomposition.

We include in the methods the following explanation:

“For some example signal visualizations, we found it useful to utilize the real-part of the wavelet transformed signal, which preserves both phase and amplitude information.”

OTHER

- lines 112-113: There are newer references to the relationship between ob lfp and respiration during active behavior

We have now included all newer references. These lines now read:

“The presence of an oscillation in the mPFC with this particular profile could also be consistent with a volume-conducted signal from the high amplitude field potentials generated by bulbar dipoles; since olfactory bulb (OB) LFP is dominated by breathing-related oscillations (Adrian 1942, Macrides et al. 1972, Fukunaga et al. 2014, Rojas-Libano et al. 2014, Ackels et al. 2020).”

- line 139: "both Ca1 PNs and INs..." confusing

We have now rephrased this sentence as follows:

“Using large-scale single-unit and laminar LFP recordings from the dorsal hippocampus, we identified that in both dorsal CA1 and dentate gyrus (DG), ~60% of PNs and 80% of CA1 INs were modulated by the phase of breathing, firing preferentially after the inspiration. . . “

- discussion: I think the work on "breathing as binder of orofacial sensation" (Kleinfeld et al 2014) may be relevant to discuss here.

We thank the reviewer for this suggestion. We have now included this very relevant aspect to the discussion.

“We suggest that centrifugal modulation by breathing is analogous to the predictive signaling employed in a wide range of neural circuits (Crapse et al. 2008), such as those underlying sensory-motor coordination (Straka et al. 2018) and likely extends to other brain structures and brain states. During active behavior, the respiratory phase modulates the processing of olfactory inputs (Shusterman et al. 2011, Jordan et al. 2018), while the coupling between breathing and whisking might underlie the coordination of active sampling processes (Moore et al. 2013), with breathing serving as the reference signal of various orofacial rhythms (Kleinfeld et al. 2014).”

REVIEWERS' COMMENTS

Reviewer #1 (Remarks to the Author):

The authors have fully addressed my concerns in the initial review and improved an already solid manuscript. I have no further concerns.

Reviewer #2 (Remarks to the Author):

Overall, the authors have addressed the questions and issues I have raised in a satisfactory manner and have thus produced a very interesting study.

If they make some minor amendments, it is absolutely fit to be published in your journal:

-Figure 7s: can they reflect on the fact that it seems that breathing-locked PFC activity paired with their induced ripples seems to be closer to the corresponding curve after olfactory deafferentiation rather than the conditions before? (quantification, results description)

-There remain some overstatements in the discussion: lines 395-395: the authors make a statement about providing a basis for mechanistic theories for information-flow through the limbic system. Yet their study looks only at one station of that system (Hippocampus, medial prefrontal cortex is not classically part of it). Please adapt the claim to avoid over interpretations.

Reviewer #3 (Remarks to the Author):

I have reviewed all the authors' responses. The authors have addressed adequately all my comments and answered all my questions.

Some of my suggestions, especially those about visualizing the variability in the data, were suggestions for the actual paper figures, and were not meant to be included as supplementary figures. But I leave that decision to the editorial staff.

I have no additional requests or comments. I think, as I expressed earlier, that the work is well-executed and it provides relevant evidence for the field. Congratulations on a great job.